# Hierarchically manufactured chiral plasmonic nanostructures with gigantic chirality for polarized emission and information encryption

Yoon Ho Lee[1,2,3,8], Yousang Won[1,8], Jungho Mun [4,8], Sanghyuk Lee[1], Yeseul Kim[4], Bongjun Yeom [5], Letian Dou [2], Junsuk Rho [4,6,7] & Joon Hak Oh [1]

Chiral metamaterials have received significant attention due to their strong chiroptical interactions with electromagnetic waves of incident light. However, the fabrication of large-area, hierarchically manufactured chiral plasmonic structures with high dissymmetry factors (*g*-factors) over a wide spectral range remains the key barrier to practical applications. Here we report a facile yet efficient method to fabricate hierarchical chiral nanostructures over a large area (>11.7 × 11.7 cm²) and with high *g*-factors (up to 0.07 in the visible region) by imparting extrinsic chirality to nanostructured polymer substrates through the simple exertion of mechanical force. We also demonstrate the application of our approach in the polarized emission of quantum dots and information encryption, including chiral quick response codes and anti-counterfeiting. This study thus paves the way for the rational design and fabrication of large-area chiral nanostructures and for their application in quantum communications and security-enhanced optical communications.

Metamaterials with multi-scale chirality can be used in various applications, including optical communication, display, biosensing, diffraction-free patterning, and chiral catalysis. This versatility mainly stems from the dissymmetric interactions between the matter and the incident electromagnetic wave, which enable the polarization rotation of linearly polarized light (LPL) and the differential absorption toward left- and right-handedness circularly polarized light (*L*-CPL and *R*-CPL). Chiral metamaterials, including chiral nanoparticles[1-6] and nanopatterns[7-14] prepared using various top-down and bottom-up approaches, exhibit unique chiroptical activities[15-17]. However, despite rapid strides in their processing architectures, critical obstacles in

relation to their scalability and uniformity persist, thereby hindering their practical applications. This is primarily attributed to the difficulty of fabricating densely arranged, large-area, high-performance metamaterials.

Although state-of-the-art chiral metamaterials exhibit remarkably enhanced chiroptical characteristics with high dissymmetry factors (*g*-factors), most studies have been limited to small-area structures (in several centimeters). The challenges related to the fabrication of complex, sophisticated nanostructures extending over a large area are the key barrier to practical applications[18-20]. For chiral nanopatterns synthesized via top-down and bottom-up approaches, a trade-off

[1]School of Chemical and Biological Engineering, Institute of Chemical Processes, Seoul National University, Seoul 08826, Republic of Korea. [2]Davidson School of Chemical Engineering, Purdue University, West Lafayette, IN 47907, USA. [3]Department of Chemistry, Purdue University, West Lafayette, IN 47907, USA. [4]Department of Mechanical Engineering, Pohang University of Science and Technology (POSTECH), Pohang 37673, Republic of Korea. [5]Department of Chemical Engineering, Hanyang University, Seoul 04763, Republic of Korea. [6]Department of Chemical Engineering, Pohang University of Science and Technology (POSTECH), Pohang 37673, Republic of Korea. [7]POSCO-POSTECH-RIST Convergence Research Center for Flat Optics and Metaphotonics, Pohang 37673, Republic of Korea. [8]These authors contributed equally: Yoon Ho Lee, Yousang Won, Jungho Mun ✉e-mail: jsrho@postech.ac.kr; joonhoh@snu.ac.kr

exists between achievable chiroptical properties and the manufacturable area[10,12,21–35]. For chiral nanoparticles synthesized via bottom-up approaches, complex and delicate fabrication techniques using chiral bio-templates are needed[1,2,5,36] and densely aligned structures are difficult to construct.

In addition, the lack of real applications for chiral plasmonic patterns is a hurdle that needs to be addressed. In the few studies on the applications of chiral metamaterials, including as CPL sensors[37–40], chiral and nonchiral materials[18,41], hologram[42,43], photoluminescence (PL) or CPL emitters[44], and quick response (QR) code[45], the chiral nanopatterns or chiral nanoparticle-dispersed films covered an area of only tens or hundreds of microns, too small for commercialization. This limitation has also posed a major roadblock for the real applications of chiral metamaterials such as the mass production of large-scale optical equipment. Moreover, circular dichroism in visible region has a wide range of applications, but it is more challenging because it requires nanostructures that are denser and smaller than those used in infrared/microwave/radio regions. Therefore, the development of a simple and cost-effective method to fabricate chiral plasmonic structures with multi-scale chirality that feature uniform nano-/micro-scale patterns over a large area is acutely desired for practical applications.

Here, we report a facile and efficient approach to construct large-area hierarchical chiral plasmonic nanostructures with high $g$-factors. We then demonstrate their application in tuning the photoluminescence (PL) of quantum dots (QDs) and in information encryption, including chiral QR codes and anti-counterfeiting technology. By imparting extrinsic chirality to nanograting polymer substrates, through the simple exertion of mechanical force and conventional metal deposition, we successfully fabricated microscopic plasmonic nanolattices with nanopatterns placed on the sloped microwrinkle surfaces (Fig. 1a). These constructions showed extrinsic chirality due to the incident oblique angle in the local nanolattice coordinate system (Fig. 1b). The hierarchical chiral plasmonic structures also exhibited a strong chiroptical activity in the visible region, with a $g$-factor >0.07 and thus 44 times higher than the values obtained without nanograting. Moreover, owing to their high plasmonic chirality, the QD-coated chiral plasmonic hybrid structures showed a high dissymmetry factor both in their PL intensities under CPL excitation ($g_{PL}$) and in their CPL-PL intensities under LPL excitation ($g_{CPL}$), 0.24 and 0.42, respectively. Furthermore, we demonstrate a simple nanoimprinting method to produce both a large-area chiral plasmonic film (11.7 × 11.7 cm²) and a chiral quick response (QR) code (1.3 × 1.3 cm²). In addition, the origin of the chiral plasmonic characteristics of the developed patterns was investigated using a comprehensive numerical analysis, and the light trapping effect of the chiral plasmonic patterns according to CPL handedness was assessed using near-field calculations. To the best of our knowledge, the area covered by the extrinsic chiral plasmonic structures is the largest (>10 × 10 cm²) described thus far and the $g$-factors (>0.07) are competitive with reported values. Our work not only provides a method to fabricate large-area chiral metamaterials with high chiroptical activities, but also has an impact on broadening the horizons for practical applications of metamaterials. These findings provide a guideline for the rational design and fabrication of large-area chiral nanostructures for use in plasmonic metamaterial applications, including optical quantum communications and information encryption.

## Results

### Fabrication and characterization of extrinsic hierarchical chiral plasmonic structures

The schematic images of hierarchical LH- and RH-chiral plasmonic structures and their corresponding optical microscopy (OM) images, showing clear mirror-image structures with chirality, are shown in Fig. 1c, d, respectively. These hierarchical chiral nanostructures, containing several nanogratings and helical micropatterns, were prepared by combining nanomolding and nanoimprinting methods with biaxial wrinkling processes. The fabrication processes of the chiral nanostructured polymer substrates and their mirror images are shown in Supplementary Fig. 1. A poly(dimethylsiloxane) (PDMS) substrate patterned with nanogratings was prepared by conventional nanomolding process using diffraction grating molds. Scanning electron microscopy (SEM) images of the nanograting-patterned PDMS/Au (40 nm) substrates, with 540 nm pitch size, are shown in Supplementary Fig. 2. The PDMS substrate with nanogratings was mechanically stretched out along one axis with UV/O₃ treatment, resulting in the formation of striped microwrinkles. Subsequently, the PDMS substrate was re-stretched, again by UV/O₃ treatment, along a direction tilted either 60° or 120° from the first axis (see the Methods for details). Through this two-step stretching process, two types of chiral microhelix wrinkled structures with mirror images with respect to the direction of the second re-stretching process were fabricated over the whole surface of the nanograting-patterned PDMS substrate, with the nanograting structures preserved on the chiral microhelical surfaces. OM images of the PDMS substrates after the first and second stretching and the corresponding schematic images are shown in Supplementary Fig. 3. The PDMS substrates with a second stretching angle of 60° or 120° clearly showed left-handed (LH) and right-handed (RH) mirror-image structures, respectively. Finally, chiral plasmonic metal nanostructures were prepared by the thermal evaporation of gold on the surface of chiral-patterned PDMS substrates.

The OM (Fig. 1c, d) and 3D profiler (Supplementary Fig. 4) images of the LH- and RH-chiral plasmonic structures clearly show that the LH- and RH-chiral plasmonic structures were well formed along the shape of the LH- and RH-patterned PDMS. The spring-like patterns had an average depth of 5 μm at an average interval of 87 μm and regularly protruding spring-ring-like patterns, with an average depth of 0.4 μm and an average spacing of 30 μm, were present along the pattern direction (Supplementary Fig. 4c, d). Not only micro-sized patterns were formed according to each stretching direction, but also nano-sized patterns that had a strong influence on chirality were additionally formed.

Figure 1e shows the enlarged OM image of the LH-patterns in which specific areas are classified in cobalt, olive and orange colors. Figure 1f–h are scanning electron microscope (SEM) images corresponding to each of the cobalt, olive, and orange color regions in Fig. 1e, respectively. The upwardly protruding part of the micropatterns (cobalt-colored area in Fig. 1e) included both the original 540 nm pitch nanogratings from the grating mold (blue dashed line) and the additionally formed 100 nm pitch nanogratings (red dashed line) perpendicular to the original nanogratings (Fig. 1f). In addition, the area around the protruding part of the micropattern (cobalt- and orange-colored area Fig. 1e) shows additional 100 nm pitch nanogratings (green dashed line) diagonal to the original nanogratings (Fig. 1g, h). As determined by atomic force microscopy (AFM), the average pitch and depth of these additionally formed nanogratings were 100 nm and 5 nm, respectively (Supplementary Fig. 5). While the grating nanopatterns with 540 nm pitch were present in all areas, the dominance of the nanopatterns additionally created in the stretching process depended on the location of the chiral micropatterns and mainly on the stretching direction applied during the formation of each region of the chiral micropattern wrinkle, because the strain direction for each step can change the orientation of each nano-wrinkle generation feature[46]. It means that all three types of nanopatterns including original (540 nm pitch), vertical (100 nm pitch), and diagonal (100 nm pitch) grating patterns exist along chiral micropatterns in the hierarchical plasmonic structures. Figure 1i, j shows the schematic images and enlarged images of the three type patterns for LH-patterns, respectively. By contrast, the nano-sized patterns were not fabricated in the flat PDMS film without the nanograting patterns (Supplementary

 

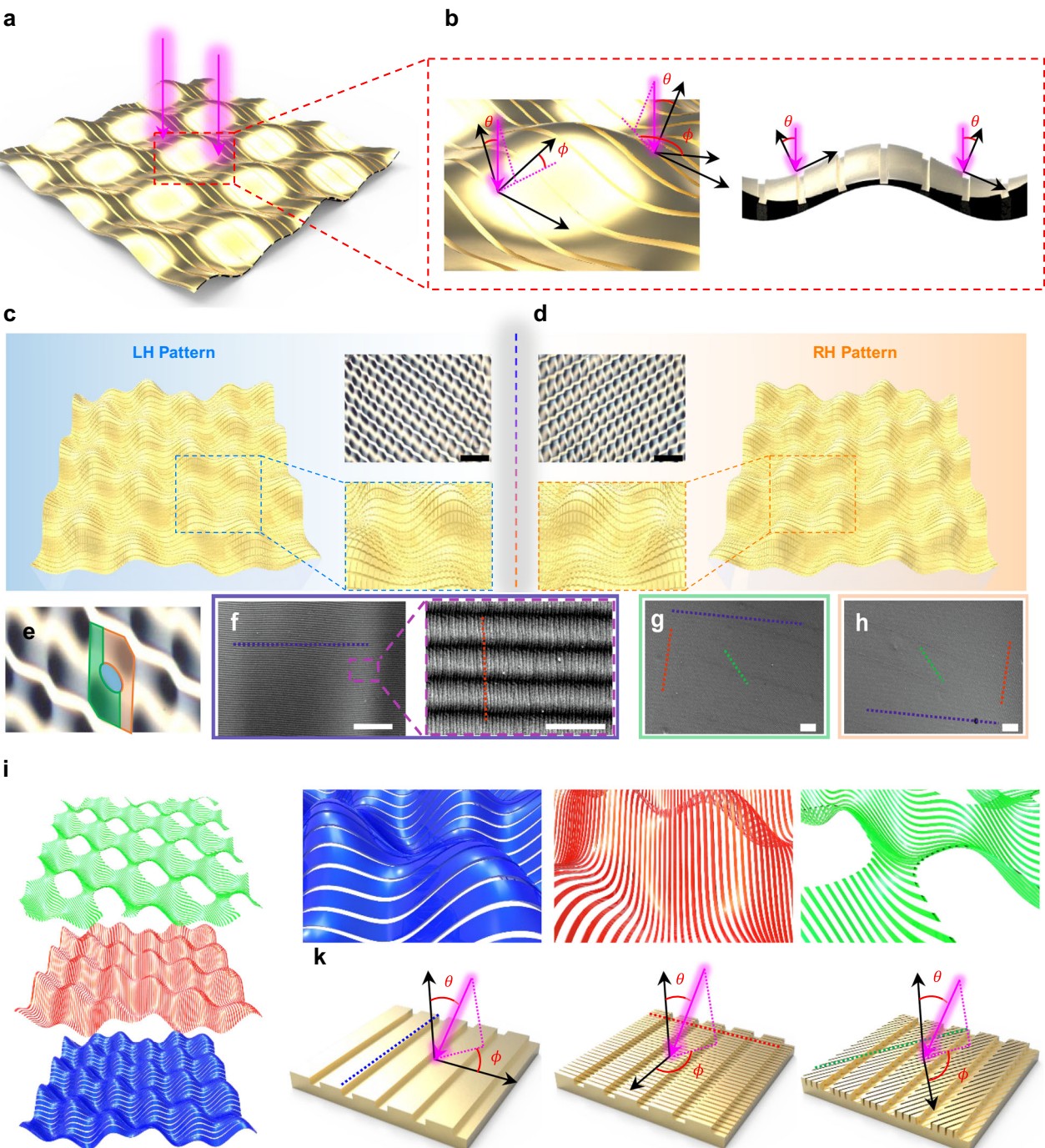

**Fig. 1 | Origin of extrinsic chiral characteristic and structure characterization of the hierarchical chiral plasmonic patterns. a, b** Schematic images of the **a** chiral plasmonic structures under light illumination. **b** Enlarged image showing the incident oblique angles ($\theta$, $\phi$) for the irradiated light. **c, d** Schematic images of hierarchical **c** LH- and **d** RH-chiral plasmonic patterns with corresponding OM images (scale bar of OM images: 50 μm). **e** Enlarged OM image of the LH-chiral plasmonic patterns, showing the central (cobalt), lower left (olive), and upper right (orange) parts of the protruding area of the micropattern (scale bar: 20 μm). **f** SEM images of the central part (cobalt area in Fig. 1e) of the protruding area of the micropattern and an enlarged image (scale bar: 10 μm and 1 μm, respectively) **g, h** SEM images of the **g** lower left (green area in Fig. 1e), and the **h** upper right (amber colored area in Fig. 1e) parts of the protruding area of the micropattern (scale bar: 2 μm). **i** Schematic images and **j** enlarged images of original (blue), vertical (red), and diagonal (green) grating patterns for LH patterns. **k** Schematic images of original grating patterns (left), additionally formed vertical nanopatterns (middle), and diagonal nanopatterns (right) for LH-chiral plasmonic patterns with $\theta$ and $\phi$ for the incident light (pink arrow: incident direction of light).

Fig. 6), indicating that the original nanograting structure plays an important role in the formation of hundreds of nanometer-sized wrinkles, by exerting the physical force needed to induce the formation of wrinkles during the contraction that follows the expansion of the PDMS substrate after the UV/O₃ stretch-release process. It has been reported that utilizing pre-patterned polymer substrates allows for the control of wrinkle direction in smaller deformations compared to the original pattern size, leveraging the role of patterns as strain relief points[47,48]. In our system, the raised step portions of the original nanogratings (540 nm pitch) may serve as strain relief points, potentially contributing to the formation of smaller hundreds of nanometer-sized wrinkles[49].

The incident obliquity angles ($\theta$ and $\phi$) of the periodic nano-patterns resulting from the different stretching directions differ from those obtained after CPL irradiation in the normal direction, which can lead to diverse extrinsic chiral plasmonic properties. Schematic images of the original, vertical, and diagonal nanogratings for LH-patterned structures (RH-patterned structures), with the $\theta$ and $\phi$ values for the incident light corresponding to each pattern, are presented in Fig. 1k (Supplementary Fig. 7). Both $\theta$ and $\phi$ differed according to the type of nanograting.

The key features of our chiral plasmonic structures are the chiral nanograting patterns and the additionally formed nanopatterns along the chiral micropatterns resulting from a simple stretching strategy, which enables extrinsic 3D chirality from the nanograting structures[50]. It is noteworthy that various types of chiral plasmonic patterns can be fabricated over a large area using different stretching directions, pitch sizes, and pattern widths of the nanogratings. Moreover, a straightforward and rapid fabrication of complex chiral meta-structures is possible by using the imprinting method that uses a patterned PDMS as the stamp[51].

## Chiroptical properties of chiral plasmonic structures

For comparison, only chiral micropatterned Au structures without nanograting patterns were prepared separately, by evaporating Au onto a chiral micropatterned PDMS substrate prepared using the same stretching process but without nanograting patterns. Moreover, in this study, inverted-chiral nanostructured polystyrene (PS) films were fabricated using the imprinting method using a chiral nanostructured PDMS substrate as the stamp by taking advantage of the fact that the complex chiral nanostructures can be simply transferred to a polymer film through a conventional imprinting method, following the procedure shown in Supplementary Fig. 8. Schematic illustrations with photographs of Au-coated (40 nm thick) chiral plasmonic patterns based on PDMS substrates with and without nanolattices are shown in Fig. 2a and Supplementary Fig. 9a. Also, images of chiral plasmonic patterns based on PS films with and without nanogratings are shown in Fig. 2b and Supplementary Fig. 9b. The inverted LH-/RH-chiral plasmonic patterns, prepared through a nano/micro imprinting process using the original LH-/RH-chiral-patterned PDMS as stamps, exhibited mirror-image shapes of the respective original LH-/RH-chiral plasmonic patterns. SEM images of inverted-LH-patterned chiral plasmonic patterns based on PS films showed the successful transfer of the chiral nanostructures of the PDMS films to the PS films (Supplementary Fig. 10). These inverted-PS/Au chiral plasmonic structures showed the inverse shape of the LH- and RH-patterned PDMS used as the stamp.

CD analyses confirmed the characteristics of the chiral plasmonic patterns. Figure 2c demonstrates the measurement of diffuse reflectance in diffuse reflectance circular dichroism (DRCD) spectroscopy, where the sample is positioned opposite the incident light entrance and the detector is placed closely to the integrating sphere at 90° from the axis of the incident light. The incident light from the left is diffusely reflected by the sample and repeatedly reflected inside the integrating sphere, finally entering the detector. Figure 2d shows the absorption spectra of the chiral plasmonic patterns. Chiral plasmonic structures with the nanogratings showed a main absorption peak at a wavelength of 350 nm with a small shoulder peak at 430 nm, while those without the nanogratings showed a main absorbance peak at 390 nm, which well corresponded to the intrinsic absorbance peak of Au[52]. Interestingly, the chiral plasmonic structures with the nanogratings showed additional shoulder peaks at wavelengths around 480 and 630 nm, presumably due to the original and additionally formed nanogratings[53,54] (Supplementary Note 1, Supplementary Figs. 11 and 12). The CD spectra showed distinctly different characteristics depending on the types of patterns[53]. The LH- and RH-patterned chiral plasmonic structures with nanogratings exhibited well-resolved bisignate Cotton effects, with vertically mirrored positive and

negative CD signals at 400 nm and 510 nm, and 625 nm, respectively (Fig. 2e). CD signals as high as >200 mdeg were detected, while the CD signals of the only chiral micropatterns were small, 5 mdeg at 530 nm. Moreover, the chiral plasmonic structures with nanogratings showed a high $g$-factor, >0.056 at a wavelength of 547 nm (Fig. 2f). This was >35 times higher than the $g$-factors determined for the only chiral micropatterns, which showed a $g$-factor of only 0.0016 in the narrow wavelength region. These results implied that the introduction of nanogratings effectively increased the chiral plasmonic properties. All CD spectra shown herein were averaged from 12 repeated measurements obtained by rotating the chiral plasmonic structures by 30° intervals to remove CD artifacts from the linear dichroism (LD) (Supplementary Fig. 13).

The CD and $g$-factor of these inverted-PS/Au chiral plasmonic structures produced vertically mirrored negative signals for LH- and RH-patterned chiral plasmonic structures, i.e., the opposite of the PDMS-based systems (Fig. 2g, h). Moreover, both the CD and $g$-factor of the PS-based chiral plasmonic patterns showed very similar values and shapes to those of the PDMS-based chiral plasmonic pattern, thus indicating that highly efficient, complex-shaped chiral plasmonic structures can be repetitively fabricated using the simple imprinting method. Furthermore, our study on the optical chirality according to the thickness of the Au and the pitch of the original nanogratings revealed that the wavelengths of the CD and the $g$-factor peaks can be controlled by changing either the thickness of the Au layer or the pitch of the nanogratings (Supplementary Note 2, Supplementary Figs. 14–16). The high $g$-factor value of 0.07, obtained at a wavelength of 700 nm using a nanograting-patterned substrate with an 830 nm pitch size, was 44 times higher than the $g$-factor value obtained with the chiral micropatterns alone.

## Theoretical analysis of hierarchical chiral plasmonic structures

That a seemingly achiral nanograting can produce chiroptical responses may seem surprising. However, those responses can be explained based on extrinsic chirality[26], i.e., asymmetric optical responses from achiral structures exposed to obliquely incident $L$-CPL and $R$-CPL. This obliquity is produced when microscopic nanograting is placed on a macroscopic slanted helical surface. To estimate the macroscopic chiral response, we first calculated the absorbance from the periodic nanograting at different obliquity angles $\theta$ and $\phi$ (Fig. 3a), or the incident angles locally experienced by the nanograting, at $L$-/$R$-CPL illuminations (see the Methods section for details). No chiral response is obtained when $\phi$ = 0°, 90°, 180°, 270°, and 360° or when $\theta$ approaches 0° (Fig. 3b) where the plane-of-incidence coincides with the mirror plane. However, a chiral response arises when the incidence plane does not lie on the mirror plane, which is the condition expected for extrinsic chirality. At $\phi$ = 45°, the absorbance at $R$-CPL is stronger than that at $L$-CPL, and the handedness with the stronger absorbance changes as $\phi$ changes by 90°, consistent with the characteristics of extrinsic chirality (Fig. 3b(i)). The maximum magnitude of CD increases as $\theta$ increases, but CD spectra have opposite signs for $\phi$ = 45° and $\phi$ = 135° (Fig. 3b(ii) and 3b(iii)), and a simple orientation averaging would result zero chiral responses as expected for extrinsic chirality. Obliquity angle dependence of absorbance for original nanopatterns with 540 nm period is given in Supplementary Fig. 17.

A macroscopic chiral response was obtained by interpolating the microscopic response. The macroscopic mapping of obliquity angles $\theta(x,y)$ and $\phi(x,y)$ for diagonal nanograting (100 nm pitch) from LH- (top panels) and RH-chiral (bottom panels) plasmonic patterns was post-processed from 3D profile measurements (Fig. 3c). The diagonal subsidiary patterns had different nanograting parameters and their distributions could be determined from $\phi' = \phi_0 + 30°$ for the LH-pattern and $\phi' = \phi_0 - 30°$ for the RH-pattern, where $\phi_0$ is the obliquity angle of the original pattern. The results clearly showed from the original (540 nm pitch) and vertical (100 nm pitch) nanogratings for the

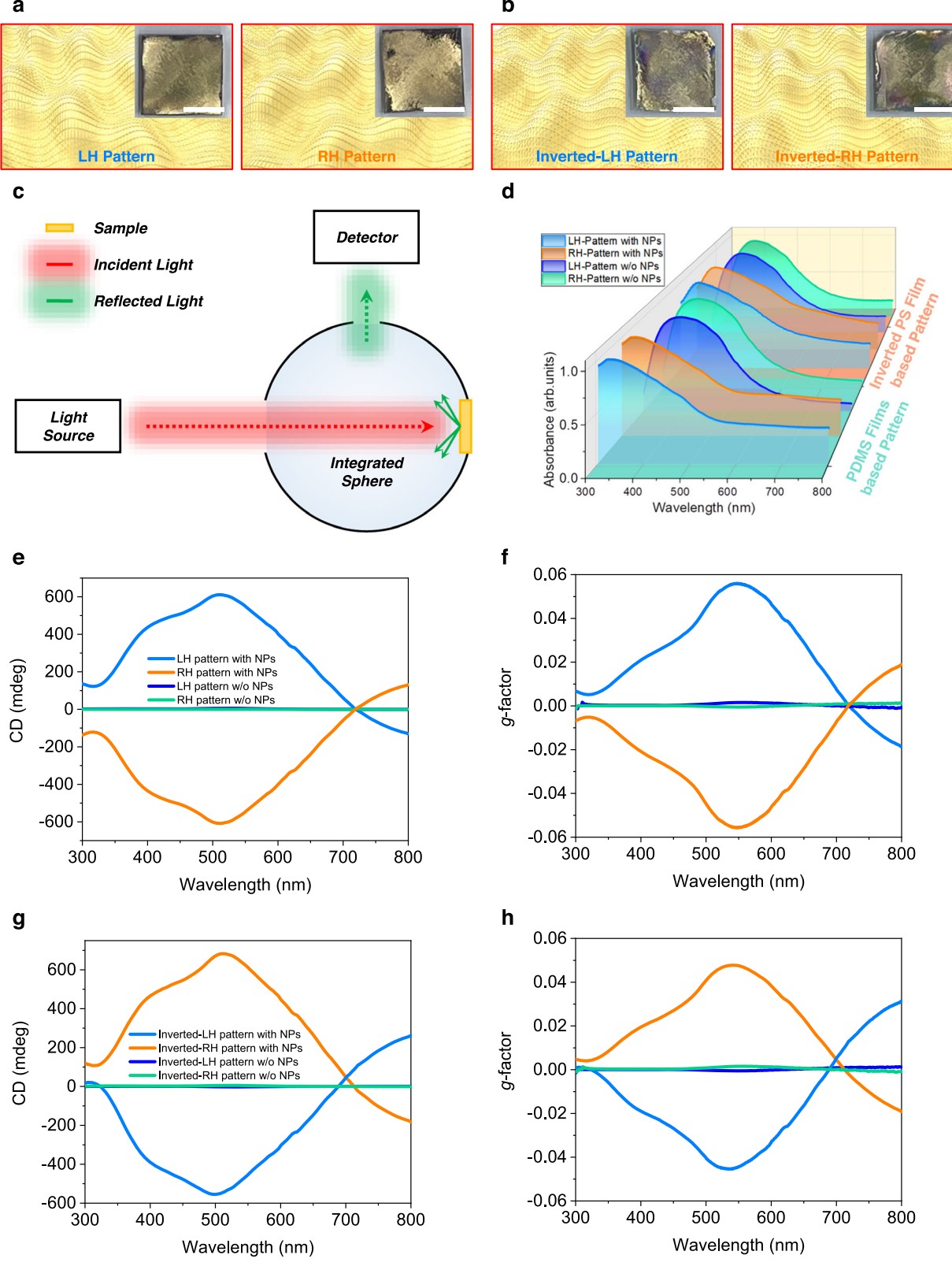

**Fig. 2 | Schematic images and optical characteristics of plasmonic chiral nanostructures based on nanograting-patterned and flat substrates.**
**a**, **b** Schematic images with corresponding OM images of **a** LH- and RH-chiral plasmonic patterns (based on PDMS films) and **b** inverted LH- and RH-chiral plasmonic patterns (based on PS films) with nanogratings (scale bars in the inset photograph images: 5 mm). **c** Schematic illustration of DRCD measurement system **d** Optical absorption spectra of chiral plasmonic patterns. **e** CD and **f** *g*-factor spectra for the chiral plasmonic patterns based on nanograting-patterned and flat PDMS substrates. **g** CD and **h** *g*-factor spectra for the inverted LH- and RH-patterned chiral plasmonic structures based on nanograting patterned and flat PS substrates.

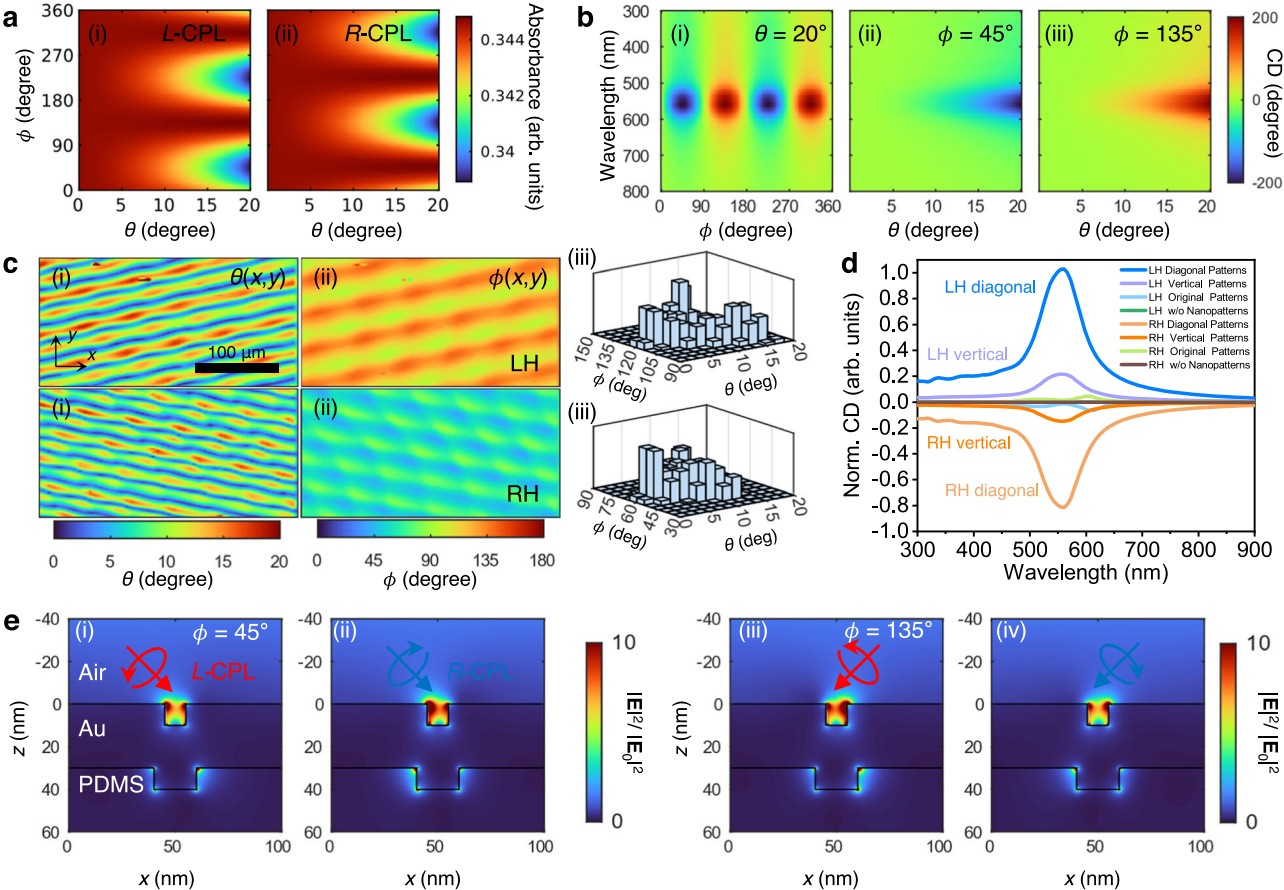

**Fig. 3 | Theoretical calculation of the optical characteristics and electric field distributions of the plasmonic chiral nanostructures. a** Obliquity angle dependence of absorbance for the diagonal nanograting under (i) *L*-CPL and (ii) *R*-CPL at 550 nm wavelength. **b** Obliquity angle dependence of CD spectra for the diagonal nanograting at different obliquity angles: (i) θ = 20°, (ii) ϕ = 45°, and (iii) ϕ = 135°. **c** Macroscopic mapping of obliquity angles (i) θ(x,y) and (ii) ϕ(x,y) and (iii) their distribution histograms for the diagonal nanograting of LH- (top panels) and RH-patterned (bottom panels) plasmonic structures. **d** Calculated CD spectra of the chiral plasmonic structures for diagonal and vertical patterns. **e** Electric field intensity distributions for the diagonal nanogratings under CPL illumination at 550 nm; (i, ii) ϕ = 45° and (iii, iv) ϕ = 135°; (i, iii) *L*-CPL and (ii, iv) *R*-CPL; θ = 30°.

LH- and RH-chiral plasmonic patterns are also shown in Supplementary Fig. 18. The obliquity mappings and histograms of vertical and original nanopatterns show that they are populated around ϕ = 0° and ϕ = 90°, respectively (Supplementary Figs. 18a(iii) and 18b(iii)), explaining their relatively weak CD signals compared to diagonal nanopatterns.

Based on the mapping, the responses were estimated by interpolating the reflection from the periodic nanogratings (see the Methods section for details). Because the macroscopic microhelical pattern was preferred over one-handedness, nonzero average chiroptical responses were obtained. The calculated normalized CD (Norm. CD) spectrum clearly showed a resonant peak near the localized surface plasmon resonance wavelength around 550 nm (Fig. 3d). The CD values of diagonal nanopatterns (100 nm pitch) were relatively higher compared to those of vertical nanopatterns (100 nm pitch), while those of the original (540 nm) nanopatterns were significantly lower than the other two nanopatterns. This approach thus provides an efficient and practical method to estimate the macroscopic optical responses of large-scale devices consisting of nanoscale microscopic patterns.

Both the experimental and the simulated CD spectra had their main peak wavelength around the localized surface plasmon resonance. However, the experimental spectrum was broader than the calculated one, because of the inhomogeneous broadening coming from the uncertainty in the parameters of the generated nanograting. In addition, the calculated CD spectra of the original patterns (540 nm pitch) differed substantially, with reversed CD signs (Supplementary Fig. 19), but the contribution from the original patterns was weaker due

to weaker plasmonic resonances of wider gap distances. The calculated model systems and calculated wavelength-dependent light absorption as a function of parameters of model are also shown in Supplementary Fig. 20.

The asymmetric chiroptical responses in the near fields were examined for LH- and RH-patterned chiral plasmonic structures by calculating the electric field distributions near the nanograting under oblique CPL illumination at 550 nm (Supplementary Fig. 21). Although the nanograting was geometrically symmetric, i.e., achiral, the obliquely incident CPLs that did not lie on the mirror planes exhibited asymmetric responses in the near fields, where the electric intensities are stronger for *R*-CPL at ϕ = 45° than for *L*-CPL at ϕ = 45° (Fig. 3e(i, ii)). Due to the symmetry, the field distribution at LCP ϕ = 45° (Fig. 3e(i)) and RCP ϕ = 135° (Fig. 3e(iv)) are inverted in the x-direction, and the same applies to RCP ϕ = 45° (Fig. 3e(ii)) and LCP ϕ = 135° (Fig. 3e(iii)). This near-field asymmetry also occurred at other wavelengths (460, 543, and 647 nm; Supplementary Fig. 21). The electric field distributions based on 540 nm pitch original patterns are also shown in Supplementary Fig. 22, indicating that the field enhancement is weaker as compared with 100 nm pitch based patterns due to the small plasmonic gap distance.

## PL characteristics of a QD/chiral plasmonic structure hybrid system

Utilizing the chiroptical characteristics of our chiral plasmonic structures with high *g*-factor values, we were able to demonstrate a

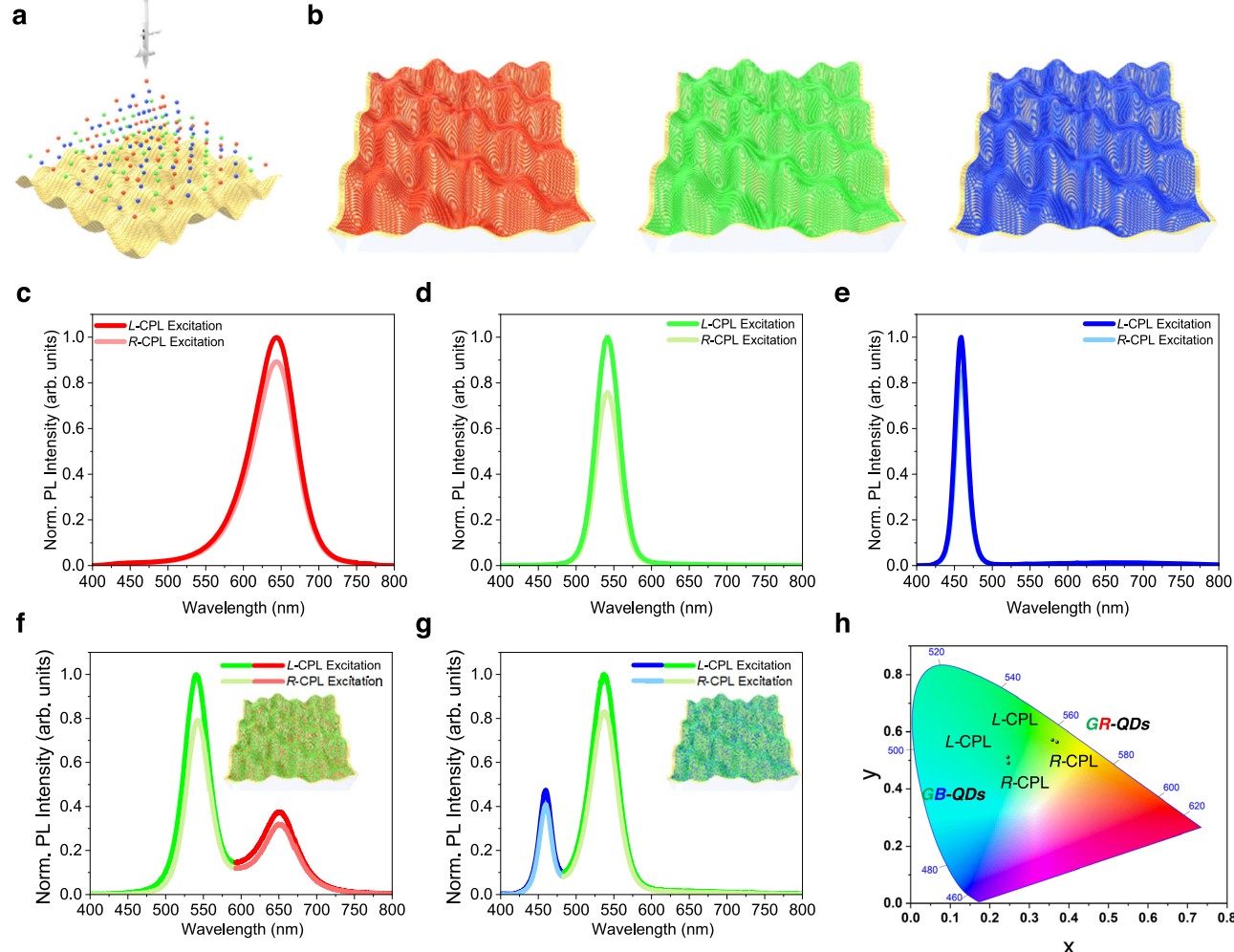

**Fig. 4 | Analyses of the PL intensity and changes in emission color of QDs according to the CPL rotation direction of the excitation light sources.** **a** Schematic image of QD spray coating on the chiral plasmonic patterns. **b** Schematic images of R-QD (left), G-QD (middle), and B-QD (right) coated LH-patterned chiral plasmonic films. **c**–**e** Polarization-sensitive PL spectra of **c** R-, **d** G-, and **e** B-QDs on the LH-patterned chiral plasmonic structures. **f**, **g** Polarization-sensitive PL spectra and schematic images of **f** GR- and **g** GB-QDs on the LH-patterned chiral plasmonic structures. **h** CIE 1931 coordinates the emitted light of the GR- and GB-QDs-based system according to CPL rotation direction.

macroscopically distinguishable change in the color of those structures, by introducing achiral QDs with PL characteristics for a specific light color according to the rotation direction of the irradiated CPL sources. The experimental setup is shown in Supplementary Fig. 23. After passing through a linear polarizer and a quarter-wave plate, the laser-generated continuous wave, at an optical wavelength of 375 nm, excited the sample. The PL signal was separated from the excitation light using a dichromic mirror and collected with a spectrometer. Red emission QD (R-QD), green emission QD (G-QD), and blue emission QD (B-QD) films were fabricated by spray coating onto the LH-patterned chiral plasmonic films (Fig. 4a). Schematic images of the R-, G-, and B-QD-coated chiral plasmonic structures are shown in Fig. 4b. The normalized PL (Norm. PL) spectra of the resulting R-, G- and B-QDs/Au-coated LH-patterned plasmonic PDMS films according to $L$-CPL and $R$-CPL irradiations are shown in Fig. 4c–e, respectively. Under CPL illumination, the PL peaks of the R-, G-, and B-QDs were located around 647, 543, and 460 nm, thus coinciding with the emission peaks of R-, G-, and B-QD, respectively. In good agreement with the CD results, these obvious PL intensity differences were observed between the $L$-CPL and $R$-CPL excitation, due to the higher absorption characteristics of $L$-CPL than $R$-CPL. Higher PL intensities of R-, G-, and B-QDs were obtained with $L$-CPL than with $R$-CPL excitation, by 11.9%, 31.8%, and 17.7%,

respectively. These results are consistent with the trend of the $g$-factor, with larger values in the order of green, blue, and red light regions. It is noted that the QDs on the flat gold substrate without chiroptical characteristics exhibited the same PL intensity in response to $L$-CPL and $R$-CPL illuminations (Supplementary Fig. 24).

To quantify the degree of PL under CPL excitation, we introduced a parameter $g_{PL}$, defined as $g_{PL} = 2(PL_{L\text{-CPL}} - PL_{R\text{-CPL}})/(PL_{L\text{-CPL}} + PL_{R\text{-CPL}})$, where $PL_{L\text{-CPL}}$ and $PL_{R\text{-CPL}}$ represent the intensity of PL under $L$-CPL and $R$-CPL excitation, respectively. The $g_{PL}$ values of the R-, G-, B-QDs were 0.10, 0.24, 0.16, respectively. Moreover, taking advantage of the higher PL intensity difference of G-QDs than R- and B-QDs depending on the CPL direction, mixed layers of G- and R-QD (GR-QDs) and of G- and B-QD (GB-QDs), both containing a larger amount of G-QDs, were prepared and spray-coated on the LH-patterned plasmonic structures. The PL intensities of the layers of GR- and GB-QD layers depending on the CPL directions are shown in Fig. 4f, g, respectively. The PL intensities of G-QDs and R-QDs for GR-QD layers under $L$-CPL excitation were 27.9% and 10% higher than those of G-QDs and R-QDs under $R$-CPL excitation, respectively, while the PL intensities of G-QDs and B-QDs for GB-QD layers under $L$-CPL excitation were 20.5% and 16.4% higher than those of G-QDs and B-QDs under $R$-CPL excitation, respectively. These results are in line with the PL intensity trends for

the single QD systems. The film mixed with R- and B-QDs (RB-QDs) also showed higher PL intensity under $L$-CPL than $R$-CPL excitation (15.3% and 11.2% higher for B-QD and R-QD, respectively; Supplementary Fig. 25). Both GR-QDs and GB-QDs showed a change in PL color along the direction of the CPL from point (0.359, 0.571) to point (0.370, 0.564) and from point (0.246, 0.511) to point (0.247, 0.490) in the CIE 1931 chromaticity space, respectively (Fig. 4h), owing to the prominent PL intensity changes from each QD type in the QD mixture films. This result demonstrates the potential of our system for use in sensing applications, based on its ability to distinguish CPL handedness through color recognition. However, the RB-QD system showed only a small PL color change along the CPL direction from point (0.199, 0.115) to point (0.204, 0.124), due to the similar PL intensity changes for R- and B-QDs (Supplementary Fig. 25c).

We then measured the direct circularly polarized emission characteristics of the hybrid system consisting of the LH-patterned plasmonic substrates and QD films. The experimental setup is shown in Supplementary Fig. 26a. After passing through a linear polarizer with an angle of 45° to surface normal, the continuous-wave laser at an optical wavelength of 405 nm excited the sample. The PL signal was collected in the forward direction and, after passing through a quarter-wave plate and a linear polarizer, was analyzed using a spectrometer. As shown in Supplementary Fig. 26b−d, there was a very strong asymmetry in the $L$-CP photoluminescence ($L$-CP-PL) and $R$-CP photoluminescence ($R$-CP-PL); this CP-PL phenomenon in the system consisting of chiral pattern and QDs may originate from the chirality transfer from extrinsic chiral systems with spin-polarized emission charateristics[55–57]. For R-, G-, and B-QDs, the $L$-CP-PL intensities were higher than the $R$-CPL-PL intensities, by ~5.8%, 52.9%, and 21.5%, respectively, consistent with the trend in the $g$-factors, with higher values in the order of green, blue, and red light regions. The intensities of the $L$- and $R$-CP-PL of QDs on the flat gold surface without chiroptical characteristics were the same (Supplementary Fig. 27). The degree of circularly polarized PL ($g_{CPL}$) for the R-, G-, B-QDs was calculated using the equation: $g_{CPL} = 2(PL_{L\text{-}CP\text{-}PL} - PL_{R\text{-}CP\text{-}PL})/(PL_{L\text{-}CP\text{-}PL} + PL_{R\text{-}CP\text{-}PL})$, where $PL_{L\text{-}CP\text{-}PL}$ and $PL_{R\text{-}CP\text{-}PL}$ represent the intensity of the $L$- and $R$-CP-PL, respectively. The resulting values of 0.06, 0.42, 0.19 implied that our chiral plasmonic structures can also impart chiroptical properties to achiral materials, further expanding the application possibilities.

### Demonstration of large-area chiral plasmonic structures and chiral QR codes

Large-area ($11.7 \times 11.7$ cm$^2$) chiral plasmonic layers were fabricated according to our simple nano-imprinting process using chiral-patterned PDMS stamp arrays to demonstrate the potential applications of our chiral patterns (Fig. 5). As shown in Supplementary Fig. 28, large-area chiral plasmonic layers were fabricated by transferring the chiral patterns of the stamp arrays to a PDMS film by conventional surface treatment and molding, with Au then deposited thereon. As seen in the SEM images of the inverted-chiral nanostructures based on PDMS films, the chiral nanostructures of the PDMS films were successfully transferred (Supplementary Fig. 29). Figure 5a shows the photograph image of inverted LH-patterned large-area ($11.7 \times 11.7$ cm$^2$) chiral plasmonic structures. To further analyze the optical uniformity of the plasmonic films, the inverted LH- and RH-patterned $11.7 \times 11.7$ cm$^2$ chiral plasmonic layers were cut into 36 $1.3 \times 1.3$ cm$^2$ pieces (Supplementary Fig. 30) and their CD spectra were measured. The 36 samples had nearly identical CD spectral characteristics and $g$-factors (Fig. 5b, c, respectively), suggesting a good uniformity of the optical properties over a large scale. Individual CD and $g$-factor spectra of these 36 chiral plasmonic pieces are also shown in Supplementary Fig. 31. The average wavelength of the CD peaks of the 36 samples was around 530 nm; the highest CD value was >700 mdeg, with narrow

standard deviations (SDs) of 3.7 nm and 71.4 mdeg, respectively (Supplementary Fig. 32). For the $g$-factors, the average wavelength of $g$-factor peaks for the 36 samples was also around 530 nm, with the highest $g$-factor value being >0.06 and narrow SDs of 4.0 nm and 0.005, respectively (Fig. 5d). These results showed that our simple method can produce high-performance chiral plasmonic patterns. It is noteworthy that our large-area chiral plasmonic patterns demonstrate the highest $g$-factor values over the largest area compared with previously reported extrinsic chiral plasmonic structures (Fig. 5e and Supplementary Table 1). The $g$-factor of our hierarchical chiral plasmonic structures was also 7.8 times higher than that previously reported for hierarchical chiral plasmonic structures based on block-copolymer-based nanopatterns[35].

In the following, we describe the production of an ultra-thin, small, chiral two-dimensional (2D) QR code in the form of a matrix representing information encryption, achieved by transferring LH- and RH- patterns to the desired area. A $1.3 \times 1.3$ cm$^2$, 54-μm thick chiral plasmonic QR code film, in which the letters "C," "H," "P," and "T" were shaped with the inverted LH-patterned chiral plasmonic structures against a background of inverted RH-patterned chiral plasmonic structures, was placed on paper money (Fig. 5f, g). The ultra-thin chiral plasmonic QD code is light enough to float on water. (Fig. 5f, right bottom). Photograph images of chiral PDMS QR code films without metal film are also shown in Supplementary Fig. 33. Chiral plasmonic QR code film showed distinctly different $g$-factor mapping values according to the pattern type (Fig. 5h). Only the letter shapes made with the inverted LH-pattern had $g$-factors of approximately −0.05, while the $g$-factors for the background area with the inverted RH-pattern was about +0.05. These proof-of-principle data clearly show that our technique can distinguish the different chiral plasmonic characteristics of a chiral QR code made up of two different chiral structures, evidence of the inherent scalability of our method. We believe that, with further development and by including a calibration procedure, this methodology can also be used for anti-counterfeiting applications, by checking the chiral optical characteristics or the direction of color coordinate changes using QDs/polymer films attached on a chiral QR code (Fig. 5i). The chirality of a QR code, which is hidden information, may also provide highly secure information and enable the inclusion of much more information in a single QR code (Fig. 5j). Both the size and pattern of the chiral QR code are easily adjustable, by controlling the size of the chiral stamp. Our method is therefore not only robust and scalable, it also paves the way for the industrial-scale production of advanced optoelectronic devices.

## Discussion

In this study, we demonstrated large-area hierarchical chiral plasmonic structures with high $g$-factors by implementing a strategy to impart extrinsic chirality to nanogratings through simple exertion of mechanical forces. Our chiral plasmonic structures showed a high $g$-factor >0.07, 44 times higher than structures without nanogratings. Moreover, QD-coated hybrid chiral plasmonic systems showed high $g_{PL}$ of 0.24 under CPL excitation and $g_{CPL}$ of 0.42 under LPL excitation. Furthermore, this method was used to successfully fabricate large-area ($11.7 \times 11.7$ cm$^2$) chiral plasmonic films and small-area ($1.3 \times 1.3$ cm$^2$) chiral QR codes. To the best of our knowledge, our chiral plasmonic patterns showed the highest $g$-factor for the largest area in comparison with other extrinsic chiral plasmonic patterns reported so far. With its potential to produce chiral plasmonic patterns of various shapes, by controlling the shapes of the grating or other nanopatterns and stretching directions, our method can be utilized for diverse applications. Insights from this study provide the theoretical guidelines needed for the design of artificial chirality and chiroptical properties for active color displays, holography, and reconfigurable chiral switches.

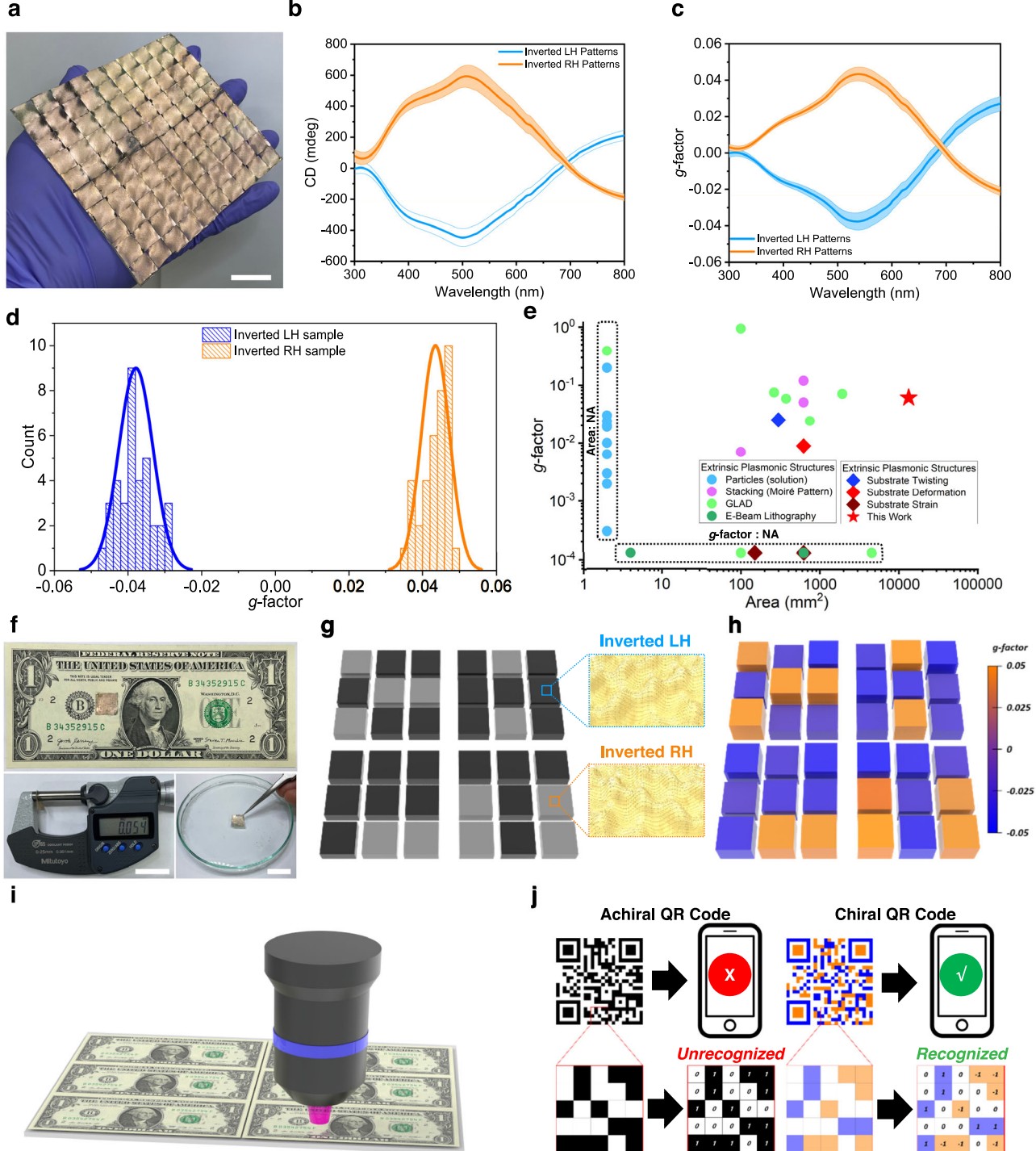

**Fig. 5 | Optical properties of the large-area (11.7 × 11.7 cm²) chiral plasmonic films and small-area (1.3 × 1.3 cm²) chiral-patterned array for QR code application. a** Photograph of the large-area inverted LH-chiral plasmonic film (scale bar: 2 cm). **b** CD and **c** g-factor spectra of the 36 pieces cut from a large area of the inverted LH- and RH- chiral plasmonic film samples. Lines and the shaded areas are mean ± SD for 36 pieces. **d** Maximum g-factor value distribution of the 36 pieces cut from a large area inverted LH- and RH-chiral plasmonic film samples. **e** Comparison of the reported chiral plasmonic patterns with those in our work with respect to the g-factor and pattern area. Related references are provided in the Supplementary

references 1–36. Related references are provided in the Supplementary Table 1 (NA: not available). **f** The chiral QR code (top) on a $1 bill: the thickness of the chiral QR code was measured using a micrometer (bottom left, scale bar: 2 cm); the chiral QR code floating in water is shown at the (bottom right, scale bar: 2 cm). **g** Schematic image, and **h** g-factor mapping results of the 54-μm-thick small-area chiral plasmonic arrays patterned in the shape of the letters "C," "H," "P," and "T" for use in anti-counterfeiting. **i, j** Schematic diagrams illustrating the potential applications of our technique in **i** anti-counterfeiting and **j** as next-generation chiral QD codes.

## Methods

### Fabrication of grating-patterned PDMS substrate
Nanograting-patterned PDMS substrates were fabricated using a conventional molding method with 184-PDMS (Sylgard, DowCorning). The PDMS precursor solution (1:10 ratio of curing agent to silicone elastomer) was poured onto a grating mold and cured at 70 °C for 2 h. Then the PDMS film was lifted off from the mold. The thickness of the grating-patterned PDMS substrates was ~250 μm.

## Fabrication of 3D chiral plasmonic-patterned chiral plasmonic film

A custom-made biaxial strain controllable stage was used to stretch the grating-patterned PDMS film. The PDMS film was placed on the stage and stretched first in the direction of the sub axis and then in the direction of the main axis. The angle between the two axes ($\theta$) was 60° for LH-patterned samples and 120° for RH-patterned samples. The strain ($\varepsilon$) along each axis was adjusted to 30%. Then the surface was modified by $O_2$ plasma treatment (Femto science, VITA) followed by a 60 min exposure to UV/$O_3$ (AhTech LTS Corp., AC-3) to form a $SiO_x$ layer on the surface of the film. Next, the uniaxial strain along the main axis was fully released to form the first wrinkles. The released film was re-stretched along the main axis to flatten it. Next, the film was exposed to UV/$O_3$ for 45 min to form thicker $SiO_2$ layers on top of the film. Then all strains were released to prepare the 3D chiral-patterned wrinkles. Finally, Au was thermally evaporated on the chiral-patterned PDMS substrate, with a deposition rate of 0.1 Å s$^{-1}$.

## Fabrication of QD-coated 3D chiral plasmonic-patterned PDMS film

Water-dispersed R-QDs, G-QDs, and B-QDs (NanoOptical Materials Inc.) were diluted with ethanol to obtain a concentration for each QD solution of 200 nmol mL$^{-1}$. The solutions were individually poured into a pot connected to a nozzle and spray-coated onto the Au-coated 3D chiral plasmonic-patterned PDMS films using an airbrush-style spray gun (IWATA HP-C plus, needle nozzle size 0.3 mm). The distance from the nozzle to the sample was 5 cm. The duration of each spraying pass was 2 s for samples with an area of 2.5 × 2.5 cm$^2$. For the G- and R-QD mixed layers (GR-QDs) and G- and B-QD mixed layers (GB-QDs), the R-QD or B-QD solution was mixed with the G-QD solution at a volume ratio of 1:2. The mixtures were spray-coated as described above.

## Fabrication of inverted 3D chiral plasmonic film by nanoimprinting process

PS solution (Aldrich, $M_n$ = 350,000, 10 wt% in toluene) was solvent-casted onto a glass Petri dish and annealed for 24 h at 70 °C under vacuum. The 3D chiral plasmonic-patterned PDMS stamp was placed in conformal contact with the PS film, which then was stamped for 24 h at 100 °C. The sample was slowly cooled to room temperature. Then the 3D chiral plasmonic-patterned PDMS mold was removed, yielding an inverted 3D chiral plasmonic-patterned PS film. Then Au (40 nm or 130 nm thick) was thermally evaporated on the chiral-patterned PS substrates at a deposition rate of 0.1 Å s$^{-1}$.

## Fabrication of an inverted large-area 3D chiral plasmonic film and a small-area chiral QR code

An inverted large-area 3D chiral plasmonic film and a small-area chiral QR code were fabricated using the nanomolding process in which a 3D chiral plasmonic PDMS film served as the master substrate mold. The 3D chiral plasmonic-patterned PDMS molds were surface-treated with n-octadecyltrichlorosilane (OTS) and placed on a glass Petri dish. For the large-area chiral plasmonic film, the PDMS molds were prepared by closely aligning 100 surface-treated chiral-patterned PDMS substrates (size: 1.2 × 1.2 cm$^2$) over an area 11.7 × 11.7 cm$^2$ in size. For the small-area chiral QR code, the PDMS molds were prepared by closely aligning 36 surface-treated LH- and RH-chiral-patterned PDMS substrates (size: 2.2 × 2.2 mm$^2$) over an area 1.3 × 1.3 cm$^2$ in size. Both the inverted large-area 3D chiral plasmonic-patterned PDMS film and the small-area chiral QD code were obtained by pouring a PDMS precursor solution (1:10 ratio of curing agent to silicone elastomer) onto the surface-treated PDMS master substrates followed by curing at 70 °C for 2 h. Then the resulting inverted large-area 3D chiral plasmonic-patterned film and small-area chiral QR code were lifted off the master substrates. The

inverted large-area 3D chiral plasmonic-patterned PDMS film had a thickness of ~250 μm. Then Au was thermally evaporated on the chiral-patterned PDMS substrate at a deposition rate of 0.1 Å s$^{-1}$ to obtain a final metal thickness of 30 nm.

## Surface characterization

SEM images were obtained using an FE-SEM (ZEISS, MERLIN Compact). OM images (Olympus, BX53M) were captured to analyze the surface of the patterned PDMS films. A 3D surface profiler (NanoFocus, μSurf) was used to measure the 3D topologies of the chiral plasmonic-patterned films. AFM images were obtained using a NX-10 AFM (Park Systems).

## Optical characterization

*CD spectra*: CD spectra were obtained using a circular dichroism spectropolarimeter equipped with an integrating sphere (JASCO, J-815 150 L) at KARA (KAIST Analysis Center for Research Advancement). *PL emissions of QDs under CPL excitation*: Samples were excited with a light source (012–63000; X-CITE 120 REPL LAMP). The filter cube contains a bandpass filter (330–385 nm) for excitation, a dichroic mirror (cutoff wavelength, 400 nm) for light splitting, and a filter (long-pass 420 nm) for emission. PL spectra were collected by a spectrometer (SpectraPro HRS-300). CPL was generated by placing an extra linear polarizer and quarter-wave plate (Thorlabs) along the pathway from the light source to the samples. The quarter-wave plate was oriented ±45° with respect to the linear polarizer. *CP-PL emissions of QDs under LPL excitation*: CPL-PL emissions of the QDs were obtained using a fluorescence spectrometer (PicoQuant FluoTime 300). Samples were excited with the spectrometer's compatible LDH-D-C-405M laser heads (exciting at 405 nm). An extra linear polarizer was placed along the pathway from the light source to the sample to make the light linearly polarize the light. The QDs on the Au-coated 3D chiral plasmonic-patterned PDMS film were photoexcited. The light emitted by the sample passed through an extra quarter-wave plate and linear polarizer along the pathway from the sample to the detector (Thorlabs). The quarter-wave plate was oriented ±45° with respect to the linear polarizer.

## Numerical simulations

The length scales of the macroscopic microhelices and microscopic nanogratings were ~30 μm and ~100 nm, respectively. Due to this large mismatch, consideration of both macroscopic and microscopic features simultaneously in numerical simulations would have been computationally intensive. Therefore, reflection from the macroscopic patterns was approximated as the sum of the reflections from the microscopic nanogratings at an oblique incidence of light (Fig. 2a). Reflection from the macroscopic patterns was obtained by interpolating the reflection from the microscopic gratings as $R_\pm = \iint \hat{R}_\pm [\theta(x,y), \phi(x,y)] w[\theta(x,y), \phi(x,y)] dx dy$, where $\hat{R}_\pm(\theta, \phi)$ is the reflection from the microscopic gratings at an oblique light incidence given by the local obliquity angles $\theta$ and $\phi$, calculated using rigorous coupled-wave analysis. Subscript ± denote L- and R-CPL. The integration domain was set as experimentally measured (Fig. 3c, d). $w(\theta, \phi)$ is the weight, and the macroscopic distribution of obliquity angles $\theta(x,y)$ and $\phi(x, y)$ was obtained using the 3D profiler. CD (in degrees) was calculated as $CD = (A_+ - A_-) \frac{\log 10}{4} \frac{180}{\pi}$, and the g-factor as $g = 2\frac{A_+ - A_-}{A_+ + A_-}$, where $A_\pm = -\log_{10} R_\pm$ is the absorbance. The optical constant of Au was taken from the tabulated results[58], and the refractive index of PDMS was set to 1.43.

## Data availability

All data are available in the main text or the supplementary materials. Raw data will be made available from the corresponding author upon request.

## Code availability

The codes used in this study are available from the corresponding author upon request.

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

## Acknowledgements

This work was supported by the Samsung Research Funding Center of Samsung Electronics under Project Number SRFC-MA1602-51. This work was also supported by the National Research Foundation of Korea (NRF) grants (NRF-2023R1A2C3007715, NRF-2021R1A4A1032515, NRF-2021M3D1A2049323, RS-2023-00281944) and Nano Material Technology Development Program (NRF-2017M3A7B8063825) funded by the Ministry of Science and ICT (MSIT) of the Korean government. This work was also supported by the Korea Toray Science Foundation. The Institute of Engineering Research at Seoul National University provided research facilities for this work. J.R. acknowledges the POSCO-POSTECH-RIST Convergence Research Center program funded by POSCO, and the NRF grants (NRF-2022M3C1A3081312, NRF-2022M3H4A1A02074314, NRF-2019R1A2C3003129, NRF-2019R1A5A8080290, RS-2023-00302586, RS-2023-00283667) funded by the MSIT of the Korean government. J.M. acknowledges the POSTECH PIURI fellowship, and the NRF *Sejong* Science fellowship (RS-2023-00252778) funded by the MSIT of the Korean government. Authors thank C. Kim (Seoul National University) for discussions, KARA (KAIST Analysis Center for Research Advancement) for the characterizations, and Y. Lee (Sbsgamedesign Co.) for help in preparing the schematic images in Supplementary Figs. 1 and 3.

## Author contributions

J.H.O., Y.H.L., and Y.W. conceived the idea. Y.H.L., Y.W., and S.H.L. fabricated and characterized the materials. J.M. and Y.K. performed the numerical simulations and analyzed the data. B.Y. helped fabrication of a hand-made stretching kit. L.D. helped with the PL measurements under CPL excitations. All authors discussed the experiments and contributed to writing the manuscript. J.R. guided the numerical simulations and optical measurements. J.H.O. guided all aspects of the work.

## Competing interests

The authors declare no competing interests
