## [Peer Review File · Nature Communications]

Hierarchically manufactured chiral plasmonic nanostructures with gigantic chirality for polarized emission and information encryptionEditorial Note: Parts of this Peer Review File have been redacted as indicated to remove third-party material where no permission to publish could be obtained.

REVIEWER COMMENTS

Reviewer #1 (Remarks to the Author):

This manuscript reports the fabrication and application of large-area plasmonic nanostructures exhibiting strong extrinsic chirality. The fabrication utilizes simple exertion of mechanical forces on nanograting polymer substrates and conventional metal deposition. The hybrid nanostructures coated with quantum dots show a high dissymmetry factor in their photoluminescence intensities. A chiral quick response (QR) code has been assessed as well for application in information encryption. In my opinion, this is interesting and comprehensive work. The manuscript is well organized and written. I recommend this manuscript for publication in Nature Communications if the authors can successfully address the following comments and questions.

1. The authors described “Interestingly, the chiral plasmonic structures with nanogratings showed additional shoulder peaks in the wavelength region ranging from 500 to 650 nm, presumably due to additionally formed nanograting and original nanogratings.” However, the additional shoulder peaks from 500 to 650 nm can hardly be observed from Fig. 2d. The authors are suggested to add simulations at least to show the peaks. Please also clarify if the main absorption peaks centered around 400 nm are due to the structure or intrinsic absorption of gold.
2. The authors mentioned “the nanograting structure plays an important role in the formation of hundreds of nanometer-sized wrinkles, by exerting the physical force needed to induce the formation of wrinkles during the contraction that follows the expansion of the PDMS substrate after the UV/O₃ stretch-release process.” The authors need to further elaborate this part with simulations/references/analyses.
3. On page 15, the authors showed that the photoluminescence of the hybrid system consisting of the plasmonic substrate and quantum dots exhibit different LCP and RCP strength. The result is interesting, since the plasmonic substrate is still achiral. If we consider

all the possible emission directions and polarization states of quantum dots, in theory, are we going to average the responses and expected to see non-chiral photoluminescence?

4. The CD signals in Fig. 2f and 2h show bisignate characteristics, but not in Fig. 3g. Please discuss the possible reason for this discrepancy.

5. It is unclear how the inverted LH-/RH-chiral plasmonic patterns look like. Please clarify it, and improve the schematics in Fig. S8.

Reviewer #2 (Remarks to the Author):

The authors describe an innovative approach for the large-scale fabrication of a chiral plasmonic material (actually, it is more of a thin film or a chiral surface). For that, they fabricate a grating in PDMS using the nanoimprint method. The PDMS film is stretched afterward twice in different directions to form crossed microwrinkles. As a part of that process, an additional grating is formed at a 100 nm length scale. After that, the sample is covered with a relatively thick gold film. The 100 nm grating offers an extrinsic chirality that can be read out at normal incidence because the grating is locally illuminated obliquely since the 100 nm gratings are placed on the microwrinkles.

The manuscript consists of four sections. First, the fabrication and characterization are discussed. Then, the properties are accessed by numerical simulations. And finally, two applications are discussed. In one, the circular emission of quantum dots is explored, and the second is an application as a chiral QR code.

The work is interesting, but relatively dense and sometimes hard to follow. However, the manuscript looks rather important to develop the field of chiral plasmonics further and enable future applications. It remains ultimately hard to judge whether the samples explored in the work will be the necessary breakthrough (in terms of size and strength of the chiral response). Still, the work is sufficiently comprehensive to work toward a publication in Nat. Commun.

Some key aspects remain unclear to me. These are:

1) How can the histogrammatic distribution of the facets, as seen by the incident field, be controlled on deterministic grounds? Or is that something that is merely observed a posteriori?

2) The comparison of the current CD signal to those of others is not clear to me. As far as I understood it, the diffused scattered light is the primary-measured quantity here. But not a single time a raw data is shown here, also compared to absorption. It remains unclear to me how meaningful the plotted CD signal is, except that it can be used to show that there is a chiral response. Please also see comment 9. Some other comments might also be a consequence of the unclear documentation in the manuscript about what is measured.

There are a few more minor comments the authors are asked to consider when working toward a revision:

3) The authors should carefully compare what has been published in the literature. For example, manuscripts such as “Large-area cavity-enhanced 3D chiral metamaterials based on the angle-dependent deposition technique” *Nanoscale*, 2020, 12, 9162 demonstrated large-scale (cm-scale) samples fabricated by self-assembly techniques offering g-factors an order of magnitude larger than those reported in the current work. Or in Reference 32, described in the manuscript as a top-down technique (which the authors of Ref. 32 clearly describe as a “scalable bottom-up approach”), cm-scale samples were obtained with dissymmetry factors as high as 0.72.

4) An unusual abbreviation such as OM should be introduced if used the first time (and not the second time).

5) There is little to see in the SEM images like Fig. S2 or 1 f, g, and h. Only with some imagination could one guess what is visible there. The quality of these images needs to be improved.

6) The record high g-factor, emphasized in the manuscript rather heavily, is only shown in a supplementary figure (S 12). Moreover, the spectrum looks strange because it seems to be

the only one that shows the spectrum only up to 700 nm, whereas all the others show the spectra up to 800 nm. Why is that? And if the highest value is shown at the edge of the displayed interval, shouldn't one expect even higher values at longer wavelengths? Does the onset of noise indicate a lack of signal, and it might worsen at longer wavelengths? But then the question would be, what exactly does such a g-factor tell the reader if no signal level is associated with it?

7) I understand that in the theory section, the scattering at the 100 nm pitch nanograting is discussed in depth. This is actually super confusing because the notion of nanograting is used twice, and I could not find any information whether the 540 nm or the 100 nm grating was considered. Only when glimpsing on the scale in Fig. 3 h and i, it looks as if the 100 nm grating was considered. Still, the imprinted grating with the lattice of 540 nm has no impact? The period compares precisely to the resonance wavelength. Have these 540 nm pitches been considered in the optical simulations, or that was considered just a part of the facet angles?

8) I understand that the 540 nm pitch nanograting structure comes from the PDMS mold, and the length scale of 87 μm comes from the stretching. But the authors also speak about a 100 nm pitch nanograting and it is important for the function (Fig. 1f). Could the authors elaborate more on the origin? The authors speak about the stretching process, but can they provide more information? I also found it confusing that the authors use the notion of nanogratings for two different kinds of structures. At one point, the distinction between the periodicities is dropped, and then it is hard to follow which structure they mean (see comment above).

9) The authors introduce in Fig. 2c diffuse reflectance circular dichroism spectroscopy, and absorption is the first result shown afterward. I cannot bring this together. How is the absorbance measured? And I am not a spectroscopic expert, but could the authors elaborate on how the CD signal is extracted from the measurement? I understand that light should not be transmitted (correct?), and the authors measure the diffusely scattered light. Do they also measure specularly reflected light to extract an actual CD signal, i.e., the difference in absorption between left- and right-handed circularly polarised light? Or which

signal is shown as the “CD signal”?

10) The absorbance as a function of ϕ looks extremely boring in the theory figures. It would potentially be more interesting to show that quantity as a function of the wavelength and θ (for a ϕ where the response is maximal) and put that ϕ dependency to the supporting information. But I am not entirely sure here.

11) Why only the normalized CD is shown in the numerics Figure 3 f? The methods section describes the calculation of the CD in degrees.

Comments from the reviewers:

Reviewer #1 (Comments to the Author):

This manuscript reports the fabrication and application of large-area plasmonic nanostructures exhibiting strong extrinsic chirality. The fabrication utilizes simple exertion of mechanical forces on nanograting polymer substrates and conventional metal deposition. The hybrid nanostructures coated with quantum dots show a high dissymmetry factor in their photoluminescence intensities. A chiral quick response (QR) code has been assessed as well for application in information encryption. In my opinion, this is interesting and comprehensive work. The manuscript is well organized and written. I recommend this manuscript for publication in Nature Communications if the authors can successfully address the following comments and questions.

Response: Thank you for your positive comments.

1. The authors described “Interestingly, the chiral plasmonic structures with nanogratings showed additional shoulder peaks in the wavelength region ranging from 500 to 650 nm, presumably due to additionally formed nanograting and original nanogratings.” However, the additional shoulder peaks from 500 to 650 nm can hardly be observed from Fig. 2d. The authors are suggested to add simulations at least to show the peaks. Please also clarify if the main absorption peaks centered around 400 nm are due to the structure or intrinsic absorption of gold.

Response: Thank you for your valuable and constructive comment. As the reviewer suggested, we conducted both experimental and simulation methods for the measurement and calculation of absorbance spectra for nanograting-patterned and flat PDMS/Au (40-nm-thick) substrates, respectively. Firstly, we would like to emphasize more clearly that the wavelength positions of the additional shoulder peaks in the optical absorption of the chiral plasmonic structure with nanogratings are at 480 nm and 630 nm. For experimental method, we measured absorbance for flat PDMS/Au, PDMS/Au with the original nanogratings, and PDMS/Au with both the original and additionally formed vertical nanogratings (original + vertical nanogratings). The absorbance spectra showed that flat PDMS/Au primarily absorbs light in the 300–500 nm range, indicating that the main absorbance peak centered around 400 nm in our chiral PDMS/Au is due to the intrinsic absorption of gold. Moreover, the absorbance spectra of the original

nanogratings PDMS/Au (as well as PDMS/Au with original + vertical nanogratings) showed higher absorption at ~480 nm wavelength compared to flat PDMS/Au. Furthermore, PDMS/Au with original + vertical nanogratings exhibited a higher absorbance peak at 480 nm than the others and an additional absorbance peak at 630 nm. The simulation results also exhibited a similar trend to the experimental results. The calculated light absorbance spectra showed that flat PDMS/Au primarily absorbs light in the 300-500 nm range, while PDMS/Au with the 100 nm pitch nanogratings, which has the same pitch size as the vertical nanogratings, exhibited an additional optical absorbance peak at 630 nm. Moreover, PDMS/Au with the nanogratings showed stronger light absorption at ~480 nm compared to the flat PDMS/Au.

The analysis could clarify that the primary absorbance peaks, centered around 400 nm, originate from the intrinsic absorption of gold, while the additional shoulder peaks around 480 and 630 nm in chiral plasmonic structures with the nanogratings are attributed to the nanogratings. Furthermore, we had planned to additionally support these explanations by citing a reference (*RSC Adv.* **2020**, *10*, 11836.), which reported the observation of peak absorbance spectra in the range of 500 to 650 nm when using Au nanogratings. However, we have identified that this reference was placed incorrectly, because it was not positioned right after the sentence following the relevant content. We have now corrected the position of the reference and included additional two references for reader's better understanding of the origin of absorbance peak in the 500-650 nm by the nanogratings (*Nanoscale*, **2017**, *9*, 1398.) and around 400 nm by the intrinsic absorption of Au layers (*Nano Energy*, **2012**, *1*, 777.). We have added the related discussion in the revised manuscript.

On page 8

Fig. 2d shows the absorbance spectra of the chiral plasmonic patterns. Chiral plasmonic structures with the nanogratings showed a main absorbance peak at a wavelength of 350 nm with a small shoulder peak at 430 nm, while those without the nanogratings showed a main absorbance peak at 390 nm, which well corresponded to the intrinsic absorbance peak of Au (*Nano Energy*, **2012**, *1*, 777.). Interestingly, the chiral plasmonic structures with the nanogratings showed additional shoulder peaks at wavelengths around 480 and 630 nm, presumably due to the original and additionally formed nanogratings (*RSC Adv.* **2020**, *10*, 11836; *Nanoscale*, **2017**, *9*, 1398.) (**Supplementary Note 1, Supplementary Fig. 11, Supplementary Fig. 12**).

In Supplementary Note 1

To investigate the origin of the light absorption peak of the chiral plasmonic structures with nanogratings, we measured absorbance spectra of three types of substrate samples: flat PDMS/Au substrates, PDMS/Au substrates with the original nanogratings, and those with original nanogratings combined with the vertical nanogratings (40-nm-thick), as shown in **Supplementary Fig. 11**. The PDMS with both the original and additionally formed vertical nanogratings was fabricated by stretching the original nanograting-patterned PDMS film perpendicular to the original nanograting direction and then applying UV/O₃ surface treatment. The schematic images of flat, original nanogratings-based, and original with additionally formed vertical nanogratings-based PDMS/Au substrates are presented in **Supplementary Fig. 11a-c**, respectively. The corresponding absorption spectra of these samples are shown in **Supplementary Fig. 11d**. The absorption spectra indicated that flat PDMS/Au predominantly absorbs light in the 300-500 nm wavelength range, confirming that the main absorption peak centered around 400 nm wavelength in our chiral PDMS/Au is due to the intrinsic absorption of Au. Additionally, both PDMS/Au substrate with the original nanogratings and PDMS/Au substrate with the original nanogratings combined with the vertical nanogratings showed increased absorption at 480 nm compared to the flat PDMS/Au substrate. Notably, the PDMS/Au substrate with the original nanogratings combined with the vertical nanogratings showed a higher absorption peak at 480 nm wavelength than the others, along with an additional absorption peak at 630 nm. It means that the additional shoulder peaks around 480 and 630 nm in chiral plasmonic structures with nanogratings are attributed to the nanogratings. The calculated absorption spectra of flat PDMS/Au substrate, PDMS/Au with the original nanogratings (540 nm pitch), and PDMS/Au with the additionally formed nanogratings (100 nm pitch) also exhibited similar trends, consistent with the experimental results of the three types of samples (**Supplementary Fig. 12**). In the flat PDMS/Au, light absorption was mainly observed in the 300-500 nm range, while the PDMS/Au with the additionally formed nanogratings (100 nm pitch) showing an additional optical absorption peak at 630 nm wavelength. Furthermore, all of the PDMS/Au with nanogratings showed stronger light

absorption at 480 nm compared to the flat PDMS/Au. The heights of the nanogratings were 10 and 20 nm for 540 nm pitch and 100 nm pitch, respectively.

For Fig. S11

Fig. S11| a-d, Schematic images of **(a)** flat PDMS/Au substrate, **(b)** PDMS/Au with the original nanogratings, **(c)** PDMS/Au substrate with the original nanogratings combined with the additionally formed vertical nanogratings, and **(d)** absorption spectra of those substrates.

For Fig. S12

Fig. S12 | Calculated absorption spectra of flat PDMS/Au substrate, PDMS/Au substrate with the original nanogratings (540 nm pitch), and PDMS/Au substrate with the additionally formed nanogratings (100 nm pitch).

2. The authors mentioned “the nanograting structure plays an important role in the formation of hundreds of nanometer-sized wrinkles, by exerting the physical force needed to induce the formation of wrinkles during the contraction that follows the expansion of the PDMS substrate after the UV/O₃ stretch-release process.” The authors need to further elaborate this part with simulations/references/analyses.

Response: We appreciate the valuable comments from the reviewer. The reviewer provides a valuable suggestion that a more detailed description is needed, regarding the reasons behind the formation of sub-hundred nanometer wrinkles during UV/O₃ treatment and substrate expansion/contraction processes. Many studies on inducing additionally formed wrinkles on the upper surface of pre-patterned PDMS have been actively reported by utilizing UV/O₃ treatment or metal deposition processes to induce compressive stress on the PDMS substrates. George M. Whitesides et al. (*Nature*, **1998**, 393, 146; *Appl. Phys. Lett.* **1999**, 75, 2557.) reported the method to control the orientation of wrinkles of PDMS substrates under applied strain conditions by using pre-patterned PDMS substrates, attributing to the significant reduction in stress owing to pre-patterning. (Fig. R1)

FIGURE REDACTED

Fig R1. (a) The stress prior to buckling near a boundary (step) of patterns in the pre-patterned PDMS substrate (*Appl. Phys. Lett.* **1999**, 75, 2557.). (b) The schematic image of grating patterned PDMS substrates (top) and the stress in the system (bottom). (c) OM image of PDMS surface after buckling. (*Nature*, **1998**, 393, 146.)

Teri W. Odom et al. (*Angew. Chem. Int. Ed.* **2014**, 53, 8117.) also reported that reducing the pattern spacing responsible for such strain relief leads to the formation of smaller wrinkle structures on the scale of nanometers. In our system, we believe that the step of the 540 nm pitch original nanogratings served as a strain relief point during the fabrication process. This strain relief might play an important role in the formation of smaller, shorter nanogratings with a pitch of 100 nm following the strain-UV/O₃ stretch-release process.

We agree with the reviewer's comment that our original description was not sufficiently detailed in explaining the cause of the formation of hundreds of nanometer-sized wrinkles. Therefore, we have additionally described this part in the revised manuscript with the citation

of these three papers (*Nature*, **1998**, 393, 146; *Appl. Phys. Lett.* **1999**, 75, 2557; *Angew. Chem. Int. Ed.* **2014**, 53, 8117.) to further elucidate this point and enhance readers' understanding.

On page 6

By contrast, the nano-sized patterns were not fabricated in the flat PDMS film without the nanograting patterns (**Supplementary Fig. 6**), indicating that the **original** nanograting structure plays an important role in the formation of hundreds of nanometer-sized wrinkles, by exerting the physical force needed to induce the formation of wrinkles during the contraction that follows the expansion of the PDMS substrate after the UV/O₃ stretch-release process. **It has been reported that utilizing pre-patterned polymer substrates allows for the control of wrinkle direction in smaller deformations compared to the original pattern size, leveraging the role of patterns as strain relief points (*Nature*, **1998**, 393, 146; *Appl. Phys. Lett.* **1999**, 75, 2557.). In our system, the raised step portions of the original nanogratings (540 nm pitch) may serve as strain relief points, potentially contributing to the formation of smaller hundreds of nanometer-sized wrinkles (*Angew. Chem. Int. Ed.* **2014**, 53, 8117.).**

On page 29

(47) Bowden, N. et al. Spontaneous formation of ordered structures in thin films of metals supported on an elastomeric polymer. *Nature* **393**, 146–149 (1998).

(48) Bowden, N. et al. The controlled formation of ordered, sinusoidal structures by plasma oxidation of an elastomeric polymer. *Appl. Phys. Lett.* **75**, 2557–2559 (1999).

(49) Huntington, M.D. et al. Controlling the Orientation of Nanowrinkles and Nanofolds by Patterning Strain in a Thin Skin Layer on a Polymer Substrate. *Angew. Chem. Int. Ed.* **53**, 8117-8121 (2014).

3. On page 15, the authors showed that the photoluminescence of the hybrid system consisting of the plasmonic substrate and quantum dots exhibit different LCP and RCP strength. The result is interesting, since the plasmonic substrate is still achiral. If we consider all the possible emission directions and polarization states of quantum dots, in theory, are we going to average the responses and expected to see non-chiral photoluminescence?

Response: The reviewer poses a thoughtful question about the possibility of theoretically non-chiral photoluminescence in a hybrid system composed of the extrinsic chiral plasmonic substrates and achiral quantum dots (QDs), where the polarization states of photoluminescent emissions in all directions are averaged. Unfortunately, due to the limitations in the current experimental set up, it is not feasible to finely control the position of the light source and photodetector in the fluorescence spectrometer (PicoQuant FluoTime 300), making it very difficult to study the effect of the emission directions. However, it is worth noting that several studies have reported selective circularly polarized photoluminescence (CP-PL) resulting from resonant coupling between the chiral plasmonic structures and the achiral light emitters, including dye (*Phys. Rev. B*, **2013**, 88, 041407; *ACS Nano*, **2016**, 10, 3389.) and QDs (*Laser Photonics Rev.* **2019**, 13, 1800276; *ACS Appl. Nano Mater.* **2019**, 2, 5681.). This resonant coupling between the chiral plasmonic structures and the achiral light emitters (QDs) leads to a difference in the transition rates of spin-up and spin-down electrons in the conduction band of the QDs to the spin-down and spin-up states in the valence band, resulting in the selective CP-PL characteristics that are independent of the direction of photoluminescent emission.

Therefore, in theory, our hybrid system is expected to exhibit chiral PL characteristics with different strengths of LCP and RCP emissions, regardless of the observation angle. There might be some variations in the PL intensity and differences in LCP and RCP emission strengths depending on the direction of PL emission due to the different angular light scattering properties of our extrinsic chiral plasmonic structures. However, these variations are not expected to be significant since QDs exhibit Lambertian light emission characteristics, resulting in minimal dependency of PL intensity on the emission direction.

4. The CD signals in Fig. 2f and 2h show bisignate characteristics, but not in Fig. 3g. Please discuss the possible reason for this discrepancy.

Response: Thank you for your valuable comments. The system consists of nanogratings with feature size (pitch) of 100~500 nm and microhelices with feature size of tens of micrometers. Brute force analysis of such multi-scale problem is extremely difficult from numerical perspectives, because the system needs to be discretized small enough to resolve the smallest nano-sized feature size but cover the entire macro-sized system. Therefore, we analyzed this multi-scaled system using a simplified model where nanogratings and microhelices with vastly different length scales are separately considered. First, the angular scattering responses of nanogratings were calculated, where chiroptical signals arise from obliquely incident lights. Then, the angular responses were mapped onto the microhelical surfaces, which effectively provide obliquity to the incident lights with respect to the nanogratings. Another simplification we utilized is that different nanograting patterns are separately considered. This simplified model would inevitably show certain discrepancies between measured and calculated results, but seems to provide quite satisfactory quantitative agreement and explains the microscopic origins of chiroptical responses from our system. Another probable reason for the discrepancy is the randomness of fabricated samples, which rely on stochastic cracking due to mechanical deformation.

5. It is unclear how the inverted LH-/RH-chiral plasmonic patterns look like. Please clarify it, and improve the schematics in Fig. S8.

Response: We thank the reviewer for the valuable comment. The inverted LH-/RH-chiral plasmonic patterns showed mirror-image shapes of the original LH-/RH-chiral plasmonic patterns owing to their fabrication through the imprinting process, where the original LH-/RH-chiral patterned PDMSs are utilized as stamps. We have added the related discussion in the revised manuscript to provide a clearer description about the shapes of the inverted LH-/RH-chiral plasmonic patterns and avoid confusing the readers. Additionally, we have prepared additional figure for the imprinting process and improved the quality of the Fig. S8 and S9 in the revised supplementary information.

On page 7

Moreover, in this study, inverted-chiral nanostructured polystyrene (PS) films were fabricated using the imprinting method using a chiral nanostructured PDMS substrate as the stamp by taking advantage of the fact that the complex chiral nanostructures can be simply transferred to a polymer film through a conventional imprinting method, following the procedure shown in **Supplementary Fig. 8**. Schematic illustrations with photographs of Au-coated (40 nm thick) chiral plasmonic patterns based on PDMS substrates with and without nanolattices are shown in **Fig. 2a** and **Supplementary Fig. 9a**. Also, images of chiral plasmonic patterns based on PS films with and without nanogratings are shown in **Fig. 2b** and **Supplementary Fig. 9b**. The inverted LH-/RH-chiral plasmonic patterns, prepared through a nano/micro imprinting process using the original LH-/RH-chiral patterned PDMS as stamps, exhibited mirror-image shapes of the respective original LH-/RH-chiral plasmonic patterns.

For Fig. S8

Fig. S8 | Schematic images of imprinting process for inverted-LH chiral patterned polymer film using LH chiral patterned PDMS stamp. The green dashed circle represents the transformation of the polymer film, located in the concave region of the LH chiral pattern PDMS stamp after nanoimprinting, into a convex area.

For Fig. S9

Fig. S9 | **a,b**, Schematic images (left) with corresponding photograph images (right) of **(a)** LH- and RH-chiral plasmonic patterns (based on PDMS films) and **(b)** inverted LH- and RH-chiral plasmonic patterns (based on PS films) without nanogratings. (red dashed arrows and circles: in the case of inverted patterns, the concave regions of the original pattern are transformed into convex regions, protruding outward, while the convex regions are transformed into concave regions, indenting inward in the opposite direction; scale bars in the photograph images: 5 mm).

We sincerely appreciate the reviewer for his/her valuable comments and suggestions on our manuscript, which were greatly helpful for improving our manuscript.

Reviewer #2

Comments:

The authors describe an innovative approach for the large-scale fabrication of a chiral plasmonic material (actually, it is more of a thin film or a chiral surface). For that, they fabricate a grating in PDMS using the nanoimprint method. The PDMS film is stretched afterward twice in different directions to form crossed microwrinkles. As a part of that process, an additional grating is formed at a 100 nm length scale. After that, the sample is covered with a relatively thick gold film. The 100 nm grating offers an extrinsic chirality that can be read out at normal incidence because the grating is locally illuminated obliquely since the 100 nm gratings are placed on the microwrinkles. The manuscript consists of four sections. First, the fabrication and characterization are discussed. Then, the properties are accessed by numerical simulations. And finally, two applications are discussed. In one, the circular emission of quantum dots is explored, and the second is an application as a chiral QR code. The work is interesting, but relatively dense and sometimes hard to follow. However, the manuscript looks rather important to develop the field of chiral plasmonics further and enable future applications. It remains ultimately hard to judge whether the samples explored in the work will be the necessary breakthrough (in terms of size and strength of the chiral response). Still, the work is sufficiently comprehensive to work toward a publication in Nat. Commun.

Response: We thank the reviewer for the valuable and positive comments.

Some key aspects remain unclear to me. These are:

1. How can the histogrammatic distribution of the facets, as seen by the incident field, be controlled on deterministic grounds? Or is that something that is merely observed a posteriori?

Response: Thank you for your thoughtful question. These results were observed a posteriori after fabricating the large area chiral plasmonic patterns. Due to the spatial limitations of the sample loading stage in CD spectra, we fabricated large area chiral plasmonic patterns and divided them into 36 pieces, matching the size of the stage. Every piece was measured individually, followed by the statistical analysis. However, the narrowness or wideness of the histogrammatic distribution depends on how uniformly inverted chiral plasmonic patterns can be fabricated on a large area through the imprinting process and Au deposition process. As the capability of the fabrication process, including imprinting and Au deposition, improves, the distribution is anticipated to become narrower. In order to fabricate these large-area chiral plasmonic patterns with good uniformity, we carried out all fabrication processes very elaborately.

2. The comparison of the current CD signal to those of others is not clear to me. As far as I understood it, the diffused scattered light is the primary-measured quantity here. But not a single time a raw data is shown here, also compared to absorption. It remains unclear to me how meaningful the plotted CD signal is, except that it can be used to show that there is a chiral response. Please also see comment 9. Some other comments might also be a consequence of the unclear documentation in the manuscript about what is measured.

Response: We appreciate your attention to detail. Below, we have described how the absorbance values for R-CPL and L-CPL are measured and CD values are calculated using the diffuse reflectance circular dichroism (DRCD) spectrometer. We believe that the following description sufficiently explains how the absorbance for R-CPL and L-CPL was measured and how CD was calculated.

The DRCD spectrometer measures light diffused or reflected from samples through preferential absorption of either incident R-CPL or L-CPL. This method was first developed by *Bilotti et al.* (*Chirality*, **2002**, *14*, 750.)

The circular dichroism (CD) spectroscopy measures the differential absorption of L-CPL and R-CPL by either chiral molecules or chiral plasmonic nanoparticles/patterns. When CPL encounters a chiral sample, the two different polarizations interact differently with the electron cloud of the sample. Due to this differential interaction, the absorption of L-CPL is different from that of R-CPL. CD spectroscopy probes this chiral property by using polarized light of opposite handedness and measuring the difference in the amount of light absorbed between the two directions of polarization. The degree of the differential absorption as a function of wavelength is called the CD spectrum. We explain the principle of CD spectroscopy based on the manual and figure provided by the CD spectroscopy instrument company (JASCO, Japan) as follows:

The CD spectroscopy measurement method is usually as follows as shown in Fig. R2:

- 1) White light, which is not polarized (that is, randomly polarized), passes through an optical filter.
- 2) Upon passing through the first prism, the white light is split into its component wavelengths, and the desired wavelength is chosen using a slit within the instrument. The prism also separates vertical and horizontal linearly polarized light (LPL). Consequently, a mix of horizontal LPL at one wavelength and vertical LPL at another wavelength passes through the slit.
- 3) In the next step, the second prism separates the contaminating vertical LPL from the horizontal LPL.
- 4) To convert LPL to CPL, the photoelastic modulator (PEM) plays an important role. During the operation of the CD spectrometer, an alternating drive voltage induces vibrations of the piezoelectric element, which in turn applies mechanical stress to the optical element, causing it to stretch and compress. The mechanical vibrations imposed on the optical element modulate its birefringence and hence the polarization state of the transmitted light.
- 5) This modulation of the optical element's birefringence leads to periodic changes in the refractive index along one axis. As a result, the component along the axis alternately

slows down and speeds up relative to the component along the other axis. The resulting phase shift between the two components of the polarized light generates CPL, with the phase shift being $\pm\pi/2$.

- 6) As R-CPL and L-CPL pass through the sample, the absorption of the two lights varies depending on the structure of the sample. This difference occurs due to the structural properties of chiral samples.
- 7) After the two CPLs pass through the sample, the detector measures the intensity of each CPL. The detector converts the difference in light intensity into an electrical signal.

FIGURE REDACTED

Fig. R2. Block diagram of a CD spectroscopy (Jasco J-810). White light is changed to horizontal linearly polarized light (LPL) by passing through the light source (LS), two prisms (P_1 and P_2), a series of mirrors (M_0 to M_5), and intermediate slits (S_1 to S_3). After being focused by a lens (L), the ordinary ray (O, horizontal LPL) produces circularly polarized light (CPL) by passing through a filter (F) and photoelastic modulator (CDM). The CPL is then passed through the shutter (SH) and the sample, and is detected by the photomultiplier (PM).

The diffuse reflectance (DR) spectroscopy measurement method is as follows:

- 1) Light incidence on sample: The DRCD directs light onto the sample and collects the reflected light using a spherical reflector and detector.
- 2) Measurement of light intensity: The intensity of light entering the sample is expressed as the sum of the intensity of absorption, reflection, transmission and scattered light. Since there is no light passing through the sample in DRCD measurement, only the reflected light is measured, and the intensity of incident light is expressed as the sum of absorption and reflection.
- 3) Measurement of reflectance: Reflectance is calculated using the following formula, where R_{∞} exhibits the % reflectance (the subscript ∞ denotes sufficient thickness of the sample layer that is thick enough to completely hide the support), I_0 represents the intensity of incident light, J is the intensity of reflected light.

$$R_{\infty} = \frac{J}{I_0}$$

- 4) Conversion to absorbance: Reflectance is a physical value that is not logically related to absorbance. Therefore, the Kubelka-Munk equation can be used to convert reflectance to absorbance, where K and S are absorption and scattering coefficients (*ChemTexts*, 2020, 6, 2.).

$$F(R_{\infty}) = \frac{(1 - R_{\infty})^2}{2R_{\infty}} = \frac{K}{S}$$

With Beer-Lambert law, the above equation is changed as follows, where A is absorbance, ϵ is absorption coefficient, l is optical path length whose value is generally 1 in diffuse reflectance spectroscopy, respectively. Therefore, we can convert reflectance to absorbance with using both Kubelka-Munk equation and Beer-lambert law.

$$F(R_{\infty}) = \frac{K}{S} = \frac{2.302\epsilon c}{S} = \frac{2.302A}{Sl} \quad (\because A = \epsilon lc)$$

As the L- and R-CPL pass through the sample, differences in absorption and reflection occur due to the chiral nature of samples and their interaction with the CPL. In general CD spectroscopy, the detector collects the transmitted light after it has been absorbed by the sample, while in DRCD spectroscopy, the detector collects light reflected from the sample's surface. Essentially, the only difference between DRCD and CD spectroscopy lies in the type of light gathered by the detector. Furthermore, the reflectance in DRCD spectroscopy is converted to absorbance using the Kubelka-Munk equation and the Beer-Lambert Law.

Regarding the importance of the CD value, in general, CD spectrum includes the difference between the absorptions of L-/R-CPL. Therefore, the CD is then written as:

$$CD = \Delta A = A_{L-CPL} - A_{R-CPL} = \Delta \varepsilon \cdot c \cdot l$$

where c is the molar concentration of the compound, l is the light path length, and $\Delta \varepsilon = \varepsilon_L - \varepsilon_R$ is the molar circular dichroism with ε_L and ε_R being the molar extinction coefficients for L- and R-CPL, respectively. CD spectrum is generally expressed in terms of a differential absorbance or ellipticity. However, CD data is usually expressed as degrees of ellipticity, θ .

$$\theta(^{\circ}) = \Delta A \cdot \frac{\ln 10}{4} \cdot \frac{180}{\pi} \approx \Delta A \cdot 32.982$$

In continuous chiral plasmonic nanopatterns, a reciprocal ($\chi = 0$) isotropic chiral ($\kappa \neq 0$) medium (that is, Pasteur media) should be modeled by the constitutive relations as follows (*J. Opt. A Pure Appl. Opt.* **11**, 11, 114003 (2009)):

$$\begin{pmatrix} \mathbf{D} \\ \mathbf{B} \end{pmatrix} = \begin{pmatrix} \varepsilon_0 \varepsilon & (\chi + i\kappa)\sqrt{\mu_0 \varepsilon_0} \\ (\chi - i\kappa)\sqrt{\mu_0 \varepsilon_0} & \mu_0 \mu \end{pmatrix} \begin{pmatrix} \mathbf{E} \\ \mathbf{H} \end{pmatrix} \rightarrow \begin{pmatrix} \mathbf{D}/\varepsilon_0 \\ c\mathbf{B} \end{pmatrix} = \begin{pmatrix} \varepsilon_r & i\kappa \\ -i\kappa & \mu_r \end{pmatrix} \begin{pmatrix} \mathbf{E} \\ \eta_0 \mathbf{H} \end{pmatrix}$$

where \mathbf{D} is the electric displacement, \mathbf{E} is the electric field, \mathbf{B} is the magnetic flux density, \mathbf{H} is the magnetic field, c is the speed of light, ε_0 is the vacuum permittivity, ε_r is the relative permittivity, ε is the permittivity of a material, μ_0 is the vacuum permeability, μ_r is the

relative permeability, μ is the permeability of a material κ is the chirality parameter, χ is the Tellegen parameter, and $\eta_0 = (\mu_0/\epsilon_0)^{1/2}$ is the vacuum wave impedance (*Light Sci. Appl.* **9**, 139, (2020); *Nano Convergence* **2**, 24, (2015)).

According to the above equation, the CD is related to κ as follows:

$$CD \propto \text{Im}(\kappa)l$$

And the ratio between CD and absorption intensity (also called g -factor), which is dimensionless, is calculated by dividing the differential absorbance of L-CPL and R-CPL by the absorbance at each wavelength:

$$g = \frac{CD}{Abs} = \frac{A_{L-CPL} - A_{R-CPL}}{\frac{1}{2}(A_{L-CPL} + A_{R-CPL})}$$

As you can see the above equation, g -factor only represents the degree of asymmetry in the absorption arising from the rotation of specific electron transitions within a mirror-image environment in CD spectroscopy while CD shows the extent of the difference between L-CPL and R-CPL.

Moreover, the CD value is associated with the molecular or structural symmetry, structure, charge distribution, and other related factors, providing important insights into the structural and functional characteristics of the chiral molecule or the chiral plasmonic nanopattern. Even with the same g -factor, the CD value may be different and vice versa. The low absorption of the chiral molecule or the chiral plasmonic nanopattern means that the interaction with CPL is very weak. For example, in the case of an ideal chiral absorption material with a g -factor value of 2, which selectively absorbs light from L-CPL and not from R-CPL, the material will consistently exhibit a g -factor value of 2, regardless of whether the light absorption from L-CPL is 0.01, 0.1, or 1. Therefore, it is necessary to consider both g -factor and CD spectra.

We believe our explanations have provided sufficient understanding of how light absorption, CD values, and g -factor values were calculated, as well as the importance of showing both CD and g -factor spectra, which conveys more comprehensive information about the chiral characteristics.

The raw data for CD and g -factor for LH- and RH-patterned chiral plasmonic PDMS films with a 40nm thickness of Au, varying with the substrate rotation angle θ , did not exhibit significant differences, as shown in the figure below. Therefore, we have presented the average results in the manuscript without showing all individual raw data points in Fig. 2.

Fig. R3. a,b, (a) CD and (b) g -factor spectra for LH- and RH-patterned chiral plasmonic PDMS films with an Au coating 40 nm thick as a function of substrate rotation angle θ :

There are a few more minor comments the authors are asked to consider when working toward a revision:

3. The authors should carefully compare what has been published in the literature. For example, manuscripts such as “Large-area cavity-enhanced 3D chiral metamaterials based on the angle-dependent deposition technique” *Nanoscale*, 2020, 12, 9162 demonstrated large-scale (cm-scale) samples fabricated by self-assembly techniques offering g -factors an order of magnitude larger than those reported in the current work. Or in Reference 32, described in the manuscript as a top-down technique (which the authors of Ref. 32 clearly describe as a “scalable bottom-up approach”), cm-scale samples were obtained with dissymmetry factors as high as 0.72.

Response: Thank you for your valuable comments and sharing the reference (*Nanoscale*, **2020**, *12*, 9162.) with us. In this study, we focused on investigating extrinsic chiral plasmonic patterns rather than intrinsic chiral plasmonic patterns. Therefore, in Figure 5e, we aimed to demonstrate the competitive extrinsic chiral plasmonic characteristics of our large area patterns by comparing their performance with previously reported extrinsic plasmonic structures.

The reference (*Nanoscale*, **2020**, *12*, 9162.) clearly exhibits a high g-factor and claims to have achieved a size of 1 cm², although it does not present any figures supporting the fabrication of a 1 cm² size (there is no supporting evidence for the 1 cm² size fabrication). However, it is important to note that this structure demonstrates intrinsic chiral plasmonic properties. Regarding reference 32 (*Nat. Mater.* **2021**, *20*, 1024.), the chiral plasmonic structures discussed in this research also exhibit intrinsic chiral plasmonic properties. Therefore, considering these two references for the section describing the performance comparison between our own extrinsic chiral plasmonic patterns and previously reported extrinsic chiral plasmonic structures appears to be less relevant.

However, we believe it is also important to present readers with information on the current performance level of intrinsic chiral plasmonic structures. Therefore, to provide readers with an overview of the performance levels reported for both intrinsic and extrinsic chiral plasmonic patterns, we had included a summary of previously reported intrinsic chiral plasmonic structures including reference 32 (*Nat. Mater.* **2021**, *20*, 1024.) in Supplementary Table 1. But we noticed that the reference you shared (*Nanoscale*, **2020**, *12*, 9162.) is missing from Fig. 5e Supplementary Table 1. Thus, the reference (*Nanoscale*, **2020**, *12*, 9162.) has been added to Fig. 5e and Supplementary Table 1.

Regarding whether reference 32 (*Nat. Mater.* **2021**, *20*, 1024.) is related on a top-down or bottom-up approach, although this paper describes it as a bottom-up approach, it utilized a laser interference lithography (top-down approach) to fabricate a nanochannel template. Then, after aligning gold nanoparticles along these nanochannels by capillarity-assisted particle assembly (bottom-up approach), the resulting substrates were stacked at an oblique angle to demonstrate the moiré chiral plasmonic structure. Therefore, despite using the bottom-up approach to align

the gold nanoparticles, we thought that this study can be considered to have stronger characteristics of a top-down approach and hence decided to classify it as a top-down approach. However, as the reviewer commented, the reference 32 (*Nat. Mater.* **2021**, *20*, 1024.) clearly is described as a bottom-up approach. We strongly agree that referring to this reference as a top-down approach could cause significant confusion for the readers. Therefore, in order to prevent such confusion, we have revised the manuscript to include a description of both top-down and bottom-up approaches in the sentence where this reference is mentioned in the introduction section.

On page 3

For chiral nanopatterns synthesized via top-down and bottom-up approaches, a trade-off exists between achievable chiroptical properties and the manufacturable area^{10,12,20-34}.

For Fig. 5

For Table S1

Ti/Ag	GLAD with Room Temperature	0.39	NA	Intrinsic	500~520	S28
PS/Ag	GLAD with PS Sphere (2 times evaporation)	0.94	100	Intrinsic	700	S29
PS/Ag	GLAD with PS Sphere and RIE Etching (3 times evaporation)	0.07	1935	Intrinsic	664	S30

4. An unusual abbreviation such as OM should be introduced if used the first time (and not the second time).

Response: We appreciate your attention to detail. We introduced the full name of the OM (optical microscope) when it was first mentioned in the revised manuscript.

On page 5

The schematic images of hierarchical LH- and RH-chiral plasmonic structures and their corresponding optical microscopy (OM) images, showing clear mirror image structures with chirality, are shown in **Figs. 1c** and **Fig. 1d**, respectively.

5. There is little to see in the SEM images like Fig. S2 or 1 f, g, and h. Only with some imagination could one guess what is visible there. The quality of these images needs to be improved.

Response: We appreciate your valuable comments. It appeared that upon PDF conversion, the resolution of the files decreased, even though we had prepared the manuscript Word file with high-resolution figures. The high-resolution images are provided below. We will upload high-resolution image files for the final publication.

For Fig. 1f

For Fig. 1g

For Fig. S2

6. The record high g -factor, emphasized in the manuscript rather heavily, is only shown in a supplementary figure (S12). Moreover, the spectrum looks strange because it seems to be the only one that shows the spectrum only up to 700 nm, whereas all the others show the spectra up to 800 nm. Why is that? And if the highest value is shown at the edge of the displayed interval, shouldn't one expect even higher values at longer wavelengths? Does the onset of noise indicates a lack of signal, and it might worsen at longer wavelengths? But then the question would be, what exactly does such a g -factor tell the reader if no signal level is associated with it?

Response: Thank you for your valuable comments that help improve the quality of our paper. First, the g -factor value of 0.07 was measured in chiral plasmonic nanostructures based on the original nanogratings with a pitch size of 830 nm, as shown in Fig. S13 (same as Fig. S16 in the revised supplementary information). We had chosen not to emphasize the high g -factor value over 0.08, which was observed in the wavelength region around 700 nm in Fig. S12, as it was considered an artefact from the baseline correction resulting in a decrease in absorbance at that region. Due to the presence of the artefact in the longer wavelength range over 700 nm, we only had shown the Fig. S12 for up to 700 nm to avoid confusion of readers. This artefact phenomenon occurs when the absolute value of the absorbance decreases and the value changes from positive to negative (or negative to positive). Even if there is the artefact, if the peak or shoulder is clearly visible before the artefact appears, the CD and g -factor spectra have often been reported because these values of these peaks or shoulders remain relatively unaffected by the artefacts (Nature, 2023, 617, 92. (Extended Data Fig. 2)).

However, we agree that these results could confuse readers, so we have conducted additional experiments to obtain more reliable results with highly similar CD and g -factor spectra. We have replaced the existing Fig. S12 with newly measured results and have incorporated this information as shown in Fig. S15 into the revised manuscript.

Supplementary Note 2

The chiral plasmonic patterns covered with 130 nm thick Au showed similar CD spectra but relatively lower values for both CD and g-factor, probably due to the decreased interaction of plasmonic modes at the Au/air and PDMS/Au interfaces (**Supplementary Figs. 15a and 15b**).

For Fig. S15

Fig. S15| a,b, (a) CD and **(b)** g-factor spectra for the chiral plasmonic PDMS films. **c,d, (c)** CD and **(d)** g-factor spectra for the inverted LH- and RH-patterned chiral plasmonic PS films (Au thickness: 130 nm)

7. I understand that in the theory section, the scattering at the 100 nm pitch nanograting is discussed in depth. This is actually super confusing because the notion of nanograting is used twice, and I could not find any information whether the 540 nm or the 100 nm grating was considered. Only when glimpsing on the scale in Fig. 3 h and i, it looks as if the 100 nm grating was considered. Still, the imprinted grating with the lattice of 540 nm has no impact? The period compares precisely to the resonance wavelength. Have these 540 nm pitches been considered in the optical simulations, or that was considered just a part of the facet angles?

Response: Thanks for the thoughtful question. In the description section of Fig. 1, the terms "original" (540 nm pitch), "vertical" (100 nm pitch), and "diagonal" (100 nm pitch) grating patterns were used to describe the different grating structures. These terms were also used to describe the content in Fig. 3. We acknowledge that this could lead to some confusion regarding the specific grating patterns that each term is referring to. To reduce such confusion, we have included additional mentions of the pitch sizes of nanogratings in the descriptive parts of Fig. 1 and Fig. 3 in the revised manuscript. We have also provided additional clarification in the revised manuscript regarding the specific grating patterns being discussed in figure captions.

An analysis of the original nanogratings (540 nm pitch) had been also conducted in the theoretical part. In addition to showing the calculated CD spectra of the chiral plasmonic structures for diagonal and vertical patterns in Fig. 3g, we also had performed calculations for the CD spectra of the original nanogratings, which are shown in original Supplementary Fig. 15. These spectra differ from the experimentally obtained results with inverted CD signs, indicating lower contributions from the original nanogratings (540 nm pitch) compared to the additionally formed nanogratings (100 nm pitch) in our extrinsic chiral plasmonic system. Through a process of increasing the alignment between experimentally measured light absorption spectra and theoretical calculation results, we were able to further optimize the model structure for calculation.

With the theoretical analysis based on this model, we were able to compare the CD intensities of diagonal, vertical, and original nanogratings more effectively. We replaced the original Fig. 3g with newly calculated CD results and included the results for original nanogratings (540 nm pitch) into the same figure. We believe that this enables readers to easily understand that the diagonal and vertical nanogratings (100 nm pitch) contribute significantly more to the chiral plasmonic characteristics of our extrinsic chiral plasmonic structures compared to the original

nanogratings (540 nm pitch). The original results in Fig. S15 were also replaced with newly calculated CD results (as shown in Fig. S19 into the revised Supplementary Information). Moreover, we replaced the original Fig. 3h and 3i with results obtained from the new model and improved the quality of Fig. 3h and 3i. Similarly, we also replaced the original Fig. S16 and S17 with results from the new model (as shown in Fig. S21 into revised Supplementary Information). Additionally, we conducted additional calculations for the E-field distribution of original nanogratings (540 nm pitch) and showed these results in Fig. S22 into revised Supplementary Information.

Furthermore, we presented additional results on the wavelength-dependent light absorption spectra based on the calculated parameters such as height, period, side, and width of the model (as shown in Fig. S20 into revised Supplementary Information). We believe that this allows readers to better understand how to control the parameters of the chiral plasmonic structure to achieve an increase or decrease in light absorption. We have added the related discussion in the revised manuscript.

On page 6

It means that all three types of nanopatterns including original (540 nm pitch), vertical (100 nm pitch), and diagonal (100 nm pitch) grating patterns exist along chiral micropatterns in the hierarchical plasmonic structures.

On page 9-11

No chiral response is obtained when $\phi = 0^\circ, 90^\circ, 180^\circ, 270^\circ,$ and 360° or when θ approaches 0° (Fig. 3b) where the plane-of-incidence coincides with the mirror plane. However, a chiral response arises when the incidence plane does not lie on the mirror plane, which is the condition expected for extrinsic chirality. At $\phi = 45^\circ$, the absorbance at *R*-CPL is stronger than that at *L*-CPL, and the handedness with the stronger absorbance changes as ϕ changes by 90° , consistent with the characteristics of extrinsic chirality (Fig. 3b(i)). The maximum magnitude of CD increases as θ increases, but CD spectra have opposite signs for $\phi = 45^\circ$ and $\phi = 135^\circ$ (Fig. 3b(ii) and 3b(iii)), and a simple orientation averaging would result zero chiral responses as

expected for extrinsic chirality. Obliquity angle dependence of absorbance for original nanopatterns with 540 nm period is given in **Supplementary Fig. 17**.

A macroscopic chiral response was obtained by interpolating the microscopic response. The macroscopic mapping of obliquity angles $\theta(x,y)$ and $\phi(x,y)$ for diagonal nanograting (100 nm pitch) from LH- (top panels) and RH-chiral (bottom panels) plasmonic patterns was post-processed from 3D profile measurements (**Fig. 3c**). The diagonal subsidiary patterns had different nanograting parameters and their distributions could be determined from $\phi' = \phi_0 + 30^\circ$ for the LH-pattern and $\phi' = \phi_0 - 30^\circ$ for the RH-pattern, where ϕ_0 is the obliquity angle of the original pattern. The results clearly showed from the original (540 nm pitch) and vertical (100 nm pitch) nanogratings for the LH- and RH-chiral plasmonic patterns are also shown in **Supplementary Fig. 18**. The obliquity mappings and histograms of vertical and original nanopatterns show that they are populated around $\phi = 0^\circ$ and $\phi = 90^\circ$, respectively (**Supplementary Figs. 18a(iii)** and **18b(iii)**), explaining their relatively weak CD signals compared to diagonal nanopatterns.

Based on the mapping, the responses were estimated by interpolating the reflection from the periodic nanogratings (see the Methods section for details). Because the macroscopic microhelical pattern was preferred over one handedness, nonzero average chiroptical responses were obtained. The obtained calculated CD spectrum clearly showed a resonant peak near the localized surface plasmon resonance wavelength around 550 nm (**Fig. 3d**). The CD values of diagonal nanopatterns (100 nm pitch) was relatively higher compared to those of vertical nanopatterns (100 nm pitch), while those of the original (540 nm) nanopatterns were significantly lower than the other two nanopatterns. This approach thus provides an efficient and practical method to estimate the macroscopic optical responses of large-scale devices consisting of nanoscale microscopic patterns.

Both the experimental and the simulated CD spectra had their main peak wavelength around the localized surface plasmon resonance. However, the experimental spectrum was broader than the calculated one, because of the inhomogeneous broadening coming from the uncertainty in the parameters of the generated nanograting. In addition, the calculated CD spectra of the original patterns (540 nm pitch) differed substantially, with reversed CD signs (**Supplementary Fig. 19**), but the contribution from the original patterns was weaker due to

weaker plasmonic resonances of wider gap distances. The calculated model systems and calculated wavelength-dependent light absorption as a function of parameters of model are also shown in **Supplementary Fig. 20**.

The asymmetric chiroptical responses in the near fields were examined for LH- and RH-patterned chiral plasmonic structures by calculating the electric field distributions near the nanograting under oblique CPL illumination at 550 nm (**Supplementary Fig. 21**). Although the nanograting was geometrically symmetric, i.e., achiral, the obliquely incident CPLs that did not lie on the mirror planes exhibited asymmetric responses in the near fields, where the electric intensities are stronger for *R*-CPL at $\phi = 45^\circ$ than for *L*-CPL at $\phi = 45^\circ$ (**Figs. 3e(i,ii)**). Due to the symmetry, the field distribution at LCP $\phi = 45^\circ$ (**Fig. 3e(i)**) and RCP $\phi = 135^\circ$ (**Fig. 3e(iv)**) are inverted in the *x*-direction, and the same applies to RCP $\phi = 45^\circ$ (**Fig. 3e(ii)**) and LCP $\phi = 135^\circ$ (**Fig. 3e(iii)**). This near field asymmetry also occurred at other wavelengths (460, 543, and 647 nm; **Supplementary Fig. 21**). The electric field distributions based on 540 nm pitch original patterns are also shown in **Supplementary Fig. 22**, indicating that the field enhancement is weaker as compared with 100 nm pitch based patterns due to the small plasmonic gap distance.

For Fig. 3

Fig. 3 | Theoretical calculation of the optical characteristics and electric field distributions of the plasmonic chiral nanostructures. (a) Obliquity angle dependence of absorbance for the diagonal nanograting under (i) *L*-CPL and (ii) *R*-CPL at 550 nm wavelength. **(b)** Obliquity angle dependence of CD spectra for the diagonal nanograting at different obliquity angles: (i) $\theta = 20^\circ$, (ii) $\phi = 45^\circ$, and (iii) $\phi = 135^\circ$. **(c)** Macroscopic mapping of obliquity angles (i) $\theta(x,y)$ and (ii) $\phi(x,y)$ and (iii) their distribution histograms for the diagonal nanograting of LH- (top panels) and RH-patterned (bottom panels) plasmonic structures. **(d)** Calculated CD spectra of the chiral plasmonic structures for diagonal and vertical patterns. **(e)** Electric field intensity distributions for the diagonal nanogratings under CPL illumination at 550 nm; (i,ii) $\phi = 45^\circ$ and (iii,iv) $\phi = 135^\circ$; (i,iii) *L*-CPL and (ii,iv) *R*-CPL; $\theta = 30^\circ$.

For Fig. S17

Fig. S17| Numerically calculated obliquity angle dependence of optical characteristics for the original nanogratings. (a) (i) Absorbance and (ii) CD spectra of the diagonal nanogratings at the obliquity angle $\theta = 10^\circ$ and different ϕ . (b) CD spectra of the diagonal nanogratings at the obliquity angles (i) $\phi = 45^\circ$ and (ii) $\phi = 135^\circ$ and different θ .

For Fig. S19

Fig. S19| Calculated CD spectra of the chiral plasmonic structures for original nanograting patterns in Fig. 3d.

For Fig. S20

Fig. S20| Parametric studies of nanogratings for different geometric parameters. (a) Schematic model of nanogratings. **b,c** Parametric studies of absorbance spectra by changing (i) width w , (ii) period a , (iii) Au side thickness s , (iv) Au thickness t , (v) grating height h , and (vi) incidence angle θ . The default parameters are: **b**, $w = 80$ nm, $a = 100$ nm, $s = 5$ nm, $t = 30$ nm, $h = 10$ nm, $\theta = 0^\circ$; and **c**, $w = 300$ nm, $a = 540$ nm, $s = 5$ nm, $t = 30$ nm, $h = 10$ nm, $\theta = 0^\circ$. The parametric studies were performed at p -polarized incident lights.

For Fig. S21

Fig. S21| Near-field distribution of the diagonal nanogratings at RGB excitation wavelengths. a-c, The electric field intensity distribution at (a) 460 nm, (b) 543 nm, (c) 647 nm, under (i,iii) *L*-CPL and (ii,iv) *R*-CPL incident lights, (i,ii) $\phi = 45^\circ$ and (iii,iv) $\phi = 135^\circ$. The calculations were performed at $\theta = 30^\circ$.

For Fig. S22

Fig. S22 | Near-field distribution of the original nanogratings at RGB excitation wavelengths. **a-c**, The electric field intensity distribution at **(a)** 460 nm, **(b)** 543 nm, **(c)** 647 nm, under **(i,iii)** *L*-CPL and **(ii,iv)** *R*-CPL incident lights, **(i,ii)** $\phi = 45^\circ$ and **(iii,iv)** $\phi = 135^\circ$. The calculations were performed at $\theta = 30^\circ$.

8. I understand that the 540 nm pitch nanograting structure comes from the PDMS mold, and the length scale of 87 μm comes from the stretching. But the authors also speak about a 100 nm pitch nanograting and it is important for the function (Fig. 1f). Could the authors elaborate more on the origin? The authors speak about the stretching process, but can they provide more information? I also found it confusing that the authors use the notion of nanogratings for two different kinds of structures. At one point, the distinction between the periodicities is dropped, and then it is hard to follow which structure they mean (see comment above).

Response: We thank the reviewer's constructive and valuable comments. The reviewer #1 also provided a valuable suggestion, emphasizing the need for a more comprehensive explanation of the origin of sub-hundred nanometer wrinkles formed during our fabrication processes in reviewer #1's 2nd comment. In response to reviewer #1's 2nd comment, information regarding the origin of the 100 nm pitch nanograting through the stretching process has been mentioned below. We believe that the description adequately addresses this issue.

Many studies on inducing additionally formed wrinkles on the upper surface of pre-patterned PDMS have been actively reported by utilizing UV/O₃ treatment or metal deposition processes to induce compressive stress on the PDMS substrates. George M. Whitesides et al. (*Nature*, **1998**, 393, 146; *Appl. Phys. Lett.* **1999**, 75, 2557.) reported the method to control the orientation of wrinkles of PDMS substrates under applied strain conditions by using pre-patterned PDMS substrates, attributing to the significant reduction in stress owing to pre-patterning. (Fig. R1)

FIGURE REDACTED

Fig R1. (a) The stress prior to buckling near a boundary (step) of patterns in the pre-patterned PDMS substrate (*Appl. Phys. Lett.* **1999**, 75, 2557.). (b) The schematic image of grating patterned PDMS substrates (top) and the stress in the system (bottom). (c) OM image of PDMS surface after buckling. (*Nature*, **1998**, 393, 146.)

Teri W. Odom et al. (*Angew. Chem. Int. Ed.* **2014**, 53, 8117.) also reported that reducing the pattern spacing responsible for such strain relief leads to the formation of smaller wrinkle structures on the scale of nanometers. In our system, we believe that the step of the 540 nm pitch original nanogratings served as a strain relief point during the fabrication process. This strain relief might play an important role in the formation of smaller, shorter nanogratings with a pitch of 100 nm following the strain-UV/O₃ stretch-release process.

We agree with the reviewer's comment that our original description was not sufficiently detailed in explaining the cause of the formation of hundreds of nanometer-sized wrinkles. Therefore, we have additionally described this part in the revised manuscript with the citation of these three papers (*Nature*, **1998**, 393, 146; *Appl. Phys. Lett.* **1999**, 75, 2557; *Angew. Chem. Int. Ed.* **2014**, 53, 8117.) to further elucidate this point and enhance readers' understanding.

Moreover, in response part to the reviewer's 7th comment, which pointed out confusion regarding the classification of nanograting types, we have addressed it in our response by mentioning the additional information about the nanograting in the revised manuscript.

On page 8

By contrast, the nano-sized patterns were not fabricated in the flat PDMS film without the nanograting patterns (**Supplementary Fig. 6**), indicating that the **original** nanograting structure plays an important role in the formation of hundreds of nanometer-sized wrinkles, by exerting the physical force needed to induce the formation of wrinkles during the contraction that follows the expansion of the PDMS substrate after the UV/O₃ stretch-release process. **It has been reported that utilizing pre-patterned polymer substrates allows for the control of wrinkle direction in smaller deformations compared to the original pattern size, leveraging the role of patterns as strain relief points (*Nature*, **1998**, 393, 146; *Appl. Phys. Lett.* **1999**, 75, 2557.). In our system, the raised step portions of the original nanogratings (540 nm pitch) may serve as strain relief points, potentially contributing to the formation of smaller hundreds of nanometer-sized wrinkles (*Angew. Chem. Int. Ed.* **2014**, 53, 8117.).**

On page 35

(47) Bowden, N. et al. Spontaneous formation of ordered structures in thin films of metals supported on an elastomeric polymer. *Nature* **393**, 146–149 (1998).

(48) Bowden, N. et al. The controlled formation of ordered, sinusoidal structures by plasma oxidation of an elastomeric polymer. *Appl. Phys. Lett.* **75**, 2557–2559 (1999).

(49) Huntington, M.D. et al. Controlling the Orientation of Nanowrinkles and Nanofolds by Patterning Strain in a Thin Skin Layer on a Polymer Substrate. *Angew. Chem. Int. Ed.* **53**, 8117-8121 (2014).

9. The authors introduce in Fig. 2c diffuse reflectance circular dichroism spectroscopy, and absorption is the first result shown afterward. I cannot bring this together. How is the absorbance measured? And I am not a spectroscopic expert, but could the authors elaborate on how the CD signal is extracted from the measurement? I understand that light should not be transmitted (correct?), and the authors measure the diffusely scattered light. Do they also measure specularly reflected light to extract an actual CD signal, i.e., the difference in absorption between left- and right-handed circularly polarised light? Or which signal is shown as the “CD signal”?

Response: Thank you for your valuable comments. In the response section of the 2nd comment, we have described how absorbance and circular dichroism (CD) are measured when using diffuse reflectance circular dichroism spectroscopy. We believe that the information provided in that section sufficiently explains the process of measuring absorbance and CD.

10. The absorbance as a function of ϕ looks extremely boring in the theory figures. It would potentially be more interesting to show that quantity as a function of the wavelength and θ (for a ϕ where the response is maximal) and put that ϕ dependency to the supporting information. But I am not entirely sure here.

Response: The reviewer has provided a valuable suggestion regarding the plotting of figures in the theoretical calculation section. Specifically, the reviewer suggests plotting the figures as a function of wavelength and θ (for ϕ where the response is maximal) and also as a function of wavelength and ϕ (for θ where the response is maximal). We really appreciate these suggestions.

The plotting design reviewer suggested would also serve as an excellent figure to explain the extrinsic chiral plasmonic properties. We have added these figures to the Fig. 3b (120 nm pitch) and Fig. S17 (540 nm pitch) with corresponding discussions in the revised manuscript to enhance the quality of the paper.

On page 9

No chiral response is obtained when $\phi = 0^\circ, 90^\circ, 180^\circ, 270^\circ,$ and 360° or when θ approaches 0° (**Fig. 3b**) where the plane-of-incidence coincides with the mirror plane. However, a chiral response arises when the incidence plane does not lie on the mirror plane, which is the condition expected for extrinsic chirality. At $\phi = 45^\circ$, the absorbance at *R*-CPL is stronger than that at *L*-CPL, and the handedness with the stronger absorbance changes as ϕ changes by 90° , consistent with the characteristics of extrinsic chirality (**Fig. 3b(i)**). The maximum magnitude of CD increases as θ increases, but CD spectra have opposite signs for $\phi = 45^\circ$ and $\phi = 135^\circ$ (**Fig. 3b(ii)** and **3b(iii)**), and a simple orientation averaging would result zero chiral responses as expected for extrinsic chirality. Obliquity angle dependence of absorbance for original nanopatterns with 540 nm period is given in **Supplementary Fig. 17**.

For Fig. 3

Fig. 3 | Theoretical calculation of the optical characteristics and electric field distributions of the plasmonic chiral nanostructures. (a) Obliquity angle dependence of absorbance for the diagonal nanograting under (i) *L*-CPL and (ii) *R*-CPL at 550 nm wavelength. **(b)** Obliquity angle dependence of CD spectra for the diagonal nanograting at different obliquity angles: (i) $\theta = 20^\circ$, (ii) $\phi = 45^\circ$, and (iii) $\phi = 135^\circ$. **(c)** Macroscopic mapping of obliquity angles (i) $\theta(x,y)$ and (ii) $\phi(x,y)$ and (iii) their distribution histograms for the diagonal nanograting of LH- (top panels) and RH-patterned (bottom panels) plasmonic structures. **(d)** Calculated CD spectra of the chiral plasmonic structures for diagonal and vertical patterns. **(e)** Electric field intensity distributions for the diagonal nanogratings under CPL illumination at 550 nm; (i,ii) $\phi = 45^\circ$ and (iii,iv) $\phi = 135^\circ$; (i,iii) *L*-CPL and (ii,iv) *R*-CPL; $\theta = 30^\circ$.

For Fig. S17

Fig. S17| Numerically calculated obliquity angle dependence of optical characteristics for the original nanogratings. (a) (i) Absorbance and (ii) CD spectra of the diagonal nanogratings at the obliquity angle $\theta = 10^\circ$ and different ϕ . (b) CD spectra of the diagonal nanogratings at the obliquity angles (i) $\phi = 45^\circ$ and (ii) $\phi = 135^\circ$ and different θ .

11. Why only the normalized CD is shown in the numerics Figure 3 f? The methods section describes the calculation of the CD in degrees.

Response: Thank you for your valuable comment. Due to the highly multi-scaled nature of this system, we relied on a simplified model to numerically analyze the origin of chiroptical responses. This simplified model successfully explained the origin of chiroptical responses in this macroscopic microhelices covered with seemingly achiral microscopic nanogratings as the extrinsic chirality arising from the nanogratings obliquely slanted with respect to the incident light. Nevertheless, the simplification would inevitably bring certain discrepancies between the measured and calculated results, limiting quantitative analysis. Therefore, we normalized the calculated CD to only qualitatively show the existence of chiroptical responses in the system.

We sincerely appreciate the reviewer for his/her valuable comments and suggestions on our manuscript, which were greatly helpful for improving our manuscript.

REVIEWERS' COMMENTS

Reviewer #1 (Remarks to the Author):

I appreciate the effort of the authors in the revision process. They have satisfactorily addressed almost all my comments, except comment 3. The substrate in this work is extrinsic chiral, and hence rigorously speaking, it is achiral. It is different from previous papers reporting selective circularly polarized photoluminescence (CP-PL) resulting from resonant coupling between the intrinsic chiral plasmonic structures and the achiral light emitters. I suggest the authors providing more in-depth analyses and discussions here.

Reviewer #2 (Remarks to the Author):

I have carefully read the response letter of the authors, and they provided entirely satisfying answers and changed the manuscript suitably. I am delighted that the authors appreciated and took the comments so seriously. The revised version significantly improved, and I do not have further comments on the paper. I suggest it to be published.

Comments from the reviewers:

Reviewer #1 (Comments to the Author):

I appreciate the effort of the authors in the revision process. They have satisfactorily addressed almost all my comments, except comment 3. The substrate in this work is extrinsic chiral, and hence rigorously speaking, it is achiral. It is different from previous papers reporting selective circularly polarized photoluminescence (CP-PL) resulting from resonant coupling between the intrinsic chiral plasmonic structures and the achiral light emitters. I suggest the authors providing more in-depth analyses and discussions here.

Response: Thank you for your positive comments.

In the manuscript, we briefly discussed that the origin of circularly polarized emission in the system with our extrinsic chiral plasmonic structures and quantum dots (QDs) is due to the coupling of the photoluminescence (PL) from the QDs with the chiral plasmonic structures. We agree with the reviewer's comment that more in-depth discussion is necessary on these phenomena occurring in extrinsic chiral systems.

First, we point out that spin-polarized scattering or emission in "achiral" systems has been reported for nanostructures [*ACS Photonics*, **2014**, *1*, 1218] and their lattices [*ACS Nano*, **2016**, *10*, 3389], where opposite circularly-polarized states scatter into different directions. Analogous to extrinsic chirality occurring in achiral systems, the averaged spin would be zero. In fact, spin-polarized emission and extrinsic chirality are reciprocal pictures. The former corresponds to near-to-far propagation of spin-polarized states into different directions, and the latter corresponds to far-to-near propagation of spin-polarized states from different directions. More recently, spin-polarized fluorescence was also reported in extrinsic chiral systems with achiral fluorescent organic molecules [*Nano Lett.* **2017**, *17*, 2265].

The CP-PL phenomenon of the system consisting of the chiral patterns and QDs may originate from the spin-polarized emission characteristics of extrinsic chiral system. In other words, (extrinsic) chirality of chiral pattern was transferred to the chiral pattern-QD system to enable CP-PL.

Therefore, to provide readers with an additional insight for overview of the principles behind the CP-PL characteristics, we have added the related discussion in the revised manuscript.

On page 12

As shown in **Supplementary Figs. 26b-d**, there was a very strong asymmetry in the *L*-CP photoluminescence (*L*-CP-PL) and *R*-CP photoluminescence (*R*-CP-PL); **this CP-PL phenomenon in the system consisting of chiral pattern and QDs may originate from the chirality transfer from extrinsic chiral systems with spin-polarized emission characteristics**

[*Laser & Photonics Reviews*, **2019**, *13*, 1800276; *Adv. Mater.* **2020**, *32*, 1907151; *Nano Lett.* **2017**, *17*, 2265].

We sincerely appreciate the reviewer for his/her valuable comments and suggestions on our manuscript, which were greatly helpful for improving our manuscript.

Comments from the reviewers:

Reviewer #2 (Comments to the Author):

I have carefully read the response letter of the authors, and they provided entirely satisfying answers and changed the manuscript suitably. I am delighted that the authors appreciated and took the comments so seriously. The revised version significantly improved, and I do not have further comments on the paper. I suggest it to be published.

Response: Thank you for your positive comments.

We sincerely appreciate the reviewer for his/her valuable comments and suggestions on our manuscript, which were greatly helpful for improving our manuscript.

REVIEWERS' COMMENTS

Reviewer #1 (Remarks to the Author):

The authors addressed my last concern, and I am happy to recommend the manuscript for publication.